# Learning Crossmodal Interaction Patterns via Attributed Bipartite Graphs for Single-Cell Omics

**Xiaotang Wang**
The Hong Kong University of Science
and Technology (Guangzhou)
xwang285@connect.hkust-gz.edu.cn

**Xuanwei Lin**
Fuzhou University
lxw_amb@foxmail.com

**Yun Zhu**
Shanghai Artificial
Intelligence Laboratory
zhuyun@pjlab.org.cn

**Hao Li**
Academy of Military
Medical Sciences
lihao_thu@163.com

**Yongqi Zhang**[*]
The Hong Kong University of Science
and Technology (Guangzhou)
yongqizhang@hkust-gz.edu.cn

## Abstract

Crossmodal matching in single-cell omics is essential for explaining biological regulatory mechanisms and enhancing downstream analyses. However, current single-cell crossmodal models often suffer from three limitations: sparse modality signals, underutilization of biological attributes, and insufficient modeling of regulatory interactions. These challenges hinder generalization in data-scarce settings and restrict the ability to uncover fine-grained biologically meaningful crossmodal relationships. Here, we present a novel framework which reformulates crossmodal matching as a graph classification task on Attributed Bipartite Graphs (ABGs). It models single-cell ATAC-RNA data as an ABG, where each expressed ATAC and RNA is treated as a distinct node with unique IDs and biological features. To model crossmodal interaction patterns on the constructed ABG, we propose Bi$^2$Former, a biologically-driven bipartite graph transformer that learns interpretable attention over ATAC–RNA pairs. This design enables the model to effectively learn and explain biological regulatory relationships between ATAC and RNA modalities. Extensive experiments demonstrate that Bi$^2$Former achieves state-of-the-art performance in crossmodal matching across diverse datasets, remains robust under sparse training data, generalizes to unseen cell types and datasets, and reveals biologically meaningful regulatory patterns. This work pioneers an ABG-based approach for single-cell crossmodal matching, offering a powerful framework for uncovering regulatory interactions at the single-cell omics. Our code is available at: https://github.com/wangxiaotang0906/Bi2Former

## 1 Introduction

Crossmodal matching [41, 40] is a fundamental task in fields such as vision-language retrieval [31], protein–description matching [12], and drug-target [13] matching, where the goal is to determine whether two modality-specific inputs correspond to the same semantic entity. This task facilitates the learning of latent interaction patterns across different data domains. In single-cell omics study, each cell is profiled with multiple modalities, such as chromatin accessibility (scATAC-seq [10]) and gene expression (scRNA-seq [30]). These modalities are inherently correlated, resulting in an intrinsic need for revealing the crossmodal interactions between them. Crossmodal matching in this context can help to identify whether a pair of ATAC and RNA profiles originates from the same cell. This

---

[*]Corresponding author.

39th Conference on Neural Information Processing Systems (NeurIPS 2025).

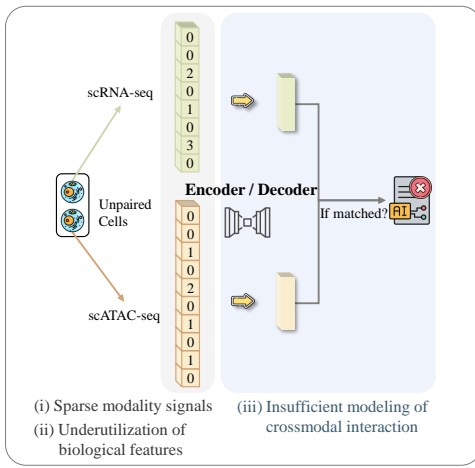
(a) Existing Challenges.

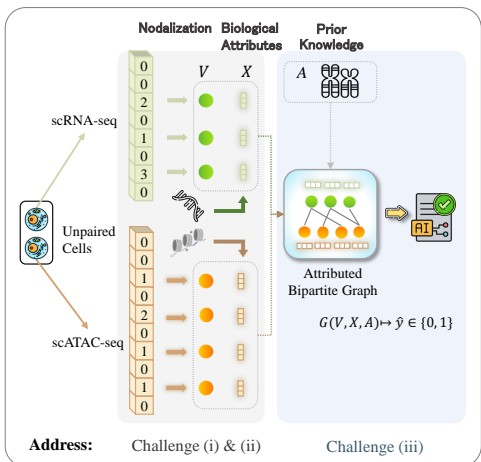
(b) Attributed Bipartite Graph Construction.

Figure 1: Comparison between (a) previous VAE-based crossmodal learning pipelines and (b) our ABG-based approach.

task offers unique opportunities to uncover regulatory mechanisms in single-cell omics, as the correct matches reflect the interaction patterns governed by underlying biology.

Existing crossmodal learning frameworks for single-cell omics, such as Cobolt [14], CLUE [35], MultiVI [3], and MIDAS [18], primarily follow a Variational Autoencoder [24] (VAE)-based architecture to perform modality alignment, data denoising, and shared latent representation learning. As shown in Figure 1a, while such methods have demonstrated efficacy in downstream tasks, they exhibit three critical limitations: (i) **Sparse modality signals**: the expression vectors of RNA and ATAC are sparse; thus, modeling them with dense vectors will introduce noise from unexpressed signals [2]. (ii) **Underutilization of biological features**: each RNA and ATAC signal carries rich biological attributes (*e.g.*, statistical summaries, genomic annotations, and DNA sequences), which are not effectively incorporated by existing methods. (iii) **Insufficient modeling of crossmodal interaction**: most current methods do not directly model the interaction between expressed ATAC and RNA signals, with limitations in understanding the underlying regulation. Moreover, most methods require paired multi-omics profiles for training, yet such data are costly and scarce. Thus leveraging limited paired data to achieve robust generalization therefore remains a major challenge.

To address these challenges, we introduce a graph-based perspective built upon the concept of Attributed Bipartite Graphs (ABGs). ABGs naturally represent two distinct node types with rich attributes and sparse interactions, making them well-suited for crossmodal biological data. This paradigm allows for explicit representation of interactions while leveraging both observed data and node-level features. Motivated by these strengths, we reformulate crossmodal matching in single-cell omics as an interaction learning problem via graph classification task on ABGs. In our setting, each expressed RNA and accessible ATAC peak is treated as a node with a unique ID and rich biological attributes, directly addressing Challenges (i) and (ii). Building on the constructed graph, we propose Bi$^2$Former, a biologically-driven bipartite graph transformer that learns interpretable attention over potential ATAC–RNA regulatory pairs, explicitly modeling crossmodal interactions (addressing Challenge (iii)). Extensive experiments show that Bi$^2$Former not only outperforms existing state-of-the-art methods in crossmodal matching accuracy across diverse datasets, but also demonstrates strong robustness under sparse training data and transfer capability across unseen cell types. Furthermore, it exhibits superior interpretability by revealing biologically meaningful regulatory patterns at both the cell-level and the cell-type-level.

Our contributions can be summarized as follows:

- To model the crossmodal data in single-cell omics which is sparse with rich attributes and important interactions, we provide a novel framework in modeling the expressed ATAC and RNA as nodes and their interactions with ABG. This establishes a valuable training corpus for single-cell omics and advances the crossmodal matching learning in this field.

- We propose Bi$^2$Former, a biologically-driven crossmodal graph transformer that integrates a biologically-driven crossmodal attention module with a message-passing architecture. This design enables the model to effectively learn and explain regulatory relationships between ATAC and RNA modalities.

- Through extensive experiments, Bi$^2$Former achieves state-of-the-art performance on crossmodal matching, including robustness under sparse training data and transfer capability across unseen cell types. Additionally, it provides superior interpretability by uncovering biologically meaningful ATAC-RNA interactions.

## 2 Related Work

### 2.1 Crossmodal Matching

Crossmodal matching aims to determine whether two modality-specific views correspond to the same underlying entity. It is a fundamental task in many domains, including vision–language retrieval [31] and audio–visual matching [9]. In the biomedical domain, crossmodal matching has been applied to problems such as protein–description matching [12], drug-target matching [13], and medical image-report retrieval [46]. These applications demonstrate the potential of crossmodal matching to uncover meaningful interactions across modalities. Typical approaches adopt dual encoders or cross-attention mechanisms to learn aligned representations across modalities.

In single-cell omics studies [37, 4], crossmodal matching, together with crossmodal generation and joint embedding, constitutes the core tasks of the field [27]. For example, CLUE [35] introduces the use of cross-encoders to construct latent representations from modality-incomplete observations. Cobolt [14] uses a shared encoder-decoder architecture to integrate multiple modalities into a unified low-dimensional latent space. MultiVI [3] extends the variational autoencoder framework to jointly model RNA and ATAC distributions through modality-specific encoders.

In this work, we explicitly focus on crossmodal matching problem as a proxy to learning interaction patterns. This formulation provides: (i) an efficient and lightweight supervision signal derived from naturally occurring cells, and (ii) aligns well with biological intuition that ATAC and RNA profiles from the same cell should reflect true regulatory interactions.

### 2.2 Learning with Attributed Bipartite Graphs

Attributed bipartite graphs (ABGs) model two distinct node types with heterogeneous features and sparse interactions, offering a powerful abstraction for many real-world problems. In recommendation systems [17], users and items are modeled as nodes in bipartite graphs, where interactions are learned through collaborative filtering [43] or Graph Neural Networks [36]. In fraud detection, ABGs have been used to represent transactional patterns between customers and merchants, capturing anomalous links through attribute-aware substructures [33]. Beyond these domains, ABGs have gained attention in biological settings for modeling drug–target or gene–disease associations [29]. Their strength lies in combining structural signals from interactions with rich semantic content at the node level.

In this work, we adopt the ABG formalism to model expressed RNA and ATAC nodes within a single cell. This allows us to filter out noise from unexpressed signals during graph construction and fully leverage the rich biological features associated with each expressed signals. In addition, the fine-grained regulatory dependencies can be captured by the proposed attention mechanism guided by biological priors over the two modalities. It is worth noting that although GLUE [6] and scMoGNN [47] also employ graph structures, they primarily rely on prior knowledge (*e.g.*, predefined guidance graphs) to facilitate multi-omics integration. By contrast, Bi$^2$Former learns and reveals regulatory knowledge rather than depending on such priors.

## 3 Graph Construction

### 3.1 Problem Definition

Given a single cell $C$, we observe two modality-specific inputs: an RNA expression vector $\mathbf{x}_{\text{RNA}} \in \mathbb{R}^{N_{\text{RNA}}}$ and an ATAC accessibility vector $\mathbf{x}_{\text{ATAC}} \in \mathbb{R}^{N_{\text{ATAC}}}$, where each element corresponds to the

expression of a gene or chromatin region. Both vectors are high-dimensional, sparse, and enriched with domain-specific annotations recorded in metadata matrices $\mathcal{H}_{\text{RNA}}$ (for RNA features) and $\mathcal{H}_{\text{ATAC}}$ (for ATAC features). The objective of crossmodal matching is to predict whether a given pair $(\mathbf{x}_{\text{RNA}}, \mathbf{x}_{\text{ATAC}})$ originates from the same biological cell. Instead of reporting soft probability scores [35, 47], we compute hard accuracy based on each pair's binary prediction. This design emphasizes precise matching signals, which are particularly important for learning fine-grained interaction patterns across modalities. A label $y \in \{0, 1\}$ is assigned to each pair, where $y = 1$ denotes a matched pair from the same cell, and $y = 0$ denotes a mismatched pair from different cells. The matched pairs are given by signals detected in the same single cell by biological experiments, while the mismatched pairs are generated by negative sampling methods. To preserve biological diversity and avoid sampling bias, negative samples are drawn proportionally according to the distribution of cell types in the dataset, and the number of positive and negative samples is maintained at a 1:1 ratio. To address the aforementioned challenges, we transform each RNA–ATAC pair into an ABG and formulate the crossmodal matching problem as a graph classification problem.

## 3.2 Graph Construction: From Modality Expression to Attributed Bipartite Graphs

Given paired single-cell expression vectors $\mathbf{x}_{\text{RNA}}$ and $\mathbf{x}_{\text{ATAC}}$, we construct a bipartite graph $G = (\mathcal{V}, X, A)$, where the node set $\mathcal{V}$ represents the expressed features in each modality, the attribute set $X$ contains the attributes of corresponding nodes, and the bipartite adjacency matrix $A$ represents the interaction between RNA nodes and ATAC nodes.

**Nodalizing Expressed Multimodal Signals into Bipartite Node Set.** The node set $\mathcal{V}$ consists of two disjoint subsets: RNA nodes and ATAC nodes. Each RNA node corresponds to a expressed gene with non-zero value in $\mathbf{x}_{\text{RNA}}$, and each ATAC node corresponds to a chromatin region with non-zero value in $\mathbf{x}_{\text{ATAC}}$. We denote them as:

$$\mathcal{V}_{\text{RNA}} = \{\text{RNA}_m \mid \mathbf{x}_{\text{RNA}}[m] \neq 0\}, \quad \mathcal{V}_{\text{ATAC}} = \{\text{ATAC}_n \mid \mathbf{x}_{\text{ATAC}}[n] \neq 0\}, \tag{1}$$

thus, the full node set of the bipartite graph is: $\mathcal{V} = \mathcal{V}_{\text{RNA}} \cup \mathcal{V}_{\text{ATAC}}$.

**Embedding Biological Attributes into Node Features.** Each node $v \in \mathcal{V}$ is associated with a biologically-informed feature vector. The overall node features are:

$$X = \{X_{\text{RNA}} \in \mathbb{R}^{|\mathcal{V}_{\text{RNA}}| \times d_r}, \ X_{\text{ATAC}} \in \mathbb{R}^{|\mathcal{V}_{\text{ATAC}}| \times d_a}\},$$

where each node feature is constructed as a concatenation:

$$X_v = \text{Concat}(\text{ID}(v), \text{Expr}(v), \text{BioAttr}(v)), \tag{2}$$

where $\text{ID}$ being a unique identity per RNA/ATAC, $\text{Expr}$ denotes the expression level of the node in the current cell, *i.e.*, the value from $\mathbf{x}_{\text{RNA}}[m]$ or $\mathbf{x}_{\text{ATAC}}[n]$, and $\text{BioAttr}$ encodes the biological metadata retrieved from $\mathcal{H}_{\text{RNA}}$ or $\mathcal{H}_{\text{ATAC}}$, such as chromosomal location, expression statistics, or DNA sequence encodings.

**Edge Design with Biological Prior Knowledge.** Edges in the constructed bipartite graph reflect potential relationships between RNA and ATAC. For edge design, a naive approach is to connect all nodes in one modality with nodes in another. However, such full connectivity introduces substantial noise because regulatory interactions are often constrained to nearby genomic loci. Alternatively, we define the adjacency matrix $A \in \{0, 1\}^{|\mathcal{V}_{\text{RNA}}| \times |\mathcal{V}_{\text{ATAC}}|}$ based on the biological prior knowledges, such as chromosomal co-location. Here, we introduce a chromosomal mask as adjacency matrix to constrain attention computation within chromosomally plausible regions. These prior-informed connections improve inductive bias and reduce noise from fully associations.

**The Constructed Attributed Bipartite Graph Formalism.** Overall, each RNA-ATAC pair is encoded as an attributed bipartite graph $G = (\mathcal{V}, X, A)$, where nodes represent expressed RNAs and ATACs with biological features, and edges reflect regulatory potential under biological priors. Thus, we reformulate the crossmodal matching problem as a binary graph classification task, where the model $f : G \mapsto \hat{y} \in \{0, 1\}$ is trained to predict if an ABG represents a matched RNA-ATAC vector.

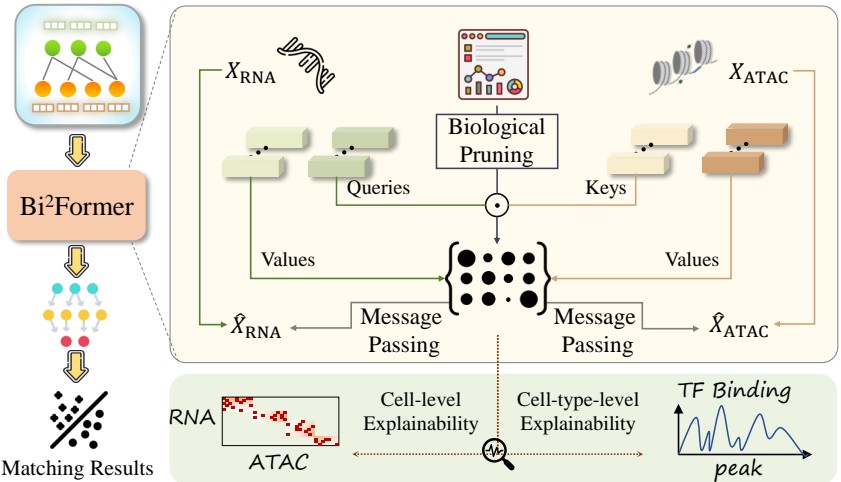

Figure 2: Model structure of Bi²Former. It consists of a biologically-driven crossmodal attention module and a crossmodal message passing architecture. Moreover, the biological pruning strategy consists of a thresholding and a top-$k$ filtering.

# 4  Bi²Former: An ABG-based Interaction Learner

To model ATAC-RNA regulatory interactions from the ABG perspective, we propose Bi²Former, a biologically-driven bipartite graph transformer that integrates a biologically-driven crossmodal attention module with a message passing architecture. Given a bipartite graph sample $G = (\mathcal{V}, X, A)$, our model processes node features and topology in two key stages:

**Biologically-driven Crossmodal Attention.**  To explicitly model the regulatory interactions between RNA and ATAC, we introduce a biologically-driven bipartitite attention mechanism. Specifically, we compute:

$$Q = X_{\text{RNA}}W_Q, \quad K = X_{\text{ATAC}}W_K, \quad V_{\text{RNA}} = X_{\text{RNA}}W_{V_r}, \quad V_{\text{ATAC}} = X_{\text{ATAC}}W_{V_a}, \tag{3}$$

where $Q, K \in \mathbb{R}^{|\mathcal{V}_{\text{RNA}}| \times d_h}$ are query and key matrices, respectively, $W_Q, W_{V_r} \in \mathbb{R}^{d_r \times d_h}$ and $W_K, W_{V_a} \in \mathbb{R}^{d_a \times d_h}$ are learnable weight matrices.

In this module, we regard the information of RNA as queries $Q$ and ATAC as keys $K$ based on the biological intuition that ATAC regions regulate RNA expression. Different from general Transformer [38] modules, we compute two separate value matrices $V_{\text{RNA}}$ and $V_{\text{ATAC}}$ for the two modalities such that the modality-specific information can be maintained. These designs can better adapt to the bipartite information in the two modalities and enable the model to explicitly learn crossmodal interaction patterns between RNA and ATAC.

As for attention values, we first compute raw crossmodal attention scores between RNA node $r$ and ATAC node $a$ using scaled dot-product over the edges defined by the adjacency matrix $A$:

$$\alpha_{r,a} = \begin{cases} \frac{Q_r K_a^\top}{\sqrt{d_h}}, & \text{if } A_{r,a} = 1 \\ 0, & \text{otherwise} \end{cases} \quad \forall (r,a) \in \mathcal{V}_{\text{RNA}} \times \mathcal{V}_{\text{ATAC}}. \tag{4}$$

To mitigate noise from under-trained nodes and suppress uniform attention distributions, we employ a biological pruning strategy. The attention scores are first passed through a sigmoid activation and thresholded by a hyperparameter $\tau$ to eliminate low-confidence signals:

$$\tilde{\alpha} = \texttt{Threshold}(\sigma(\alpha), \tau) \in \mathbb{R}^{|\mathcal{V}_{\text{RNA}}| \times |\mathcal{V}_{\text{ATAC}}|}. \tag{5}$$

Subsequently, we retain only the top-$k$ attention scores aligned with each RNA node and binarize the scores, yielding a sparse binary attention mask $\tilde{\alpha} \in \{0,1\}^{|\mathcal{V}_{\text{RNA}}| \times |\mathcal{V}_{\text{ATAC}}|}$. This constraint aligns with the biological truth that each gene is typically regulated by a limited number of ATAC peaks [15]. The resulting binary attention matrix $\tilde{\alpha}$ serves both as an interpretable crossmodal regulatory map (see details in Appendix C) and as a structured graph to guide the subsequent message passing stage.

**Crossmodal Message Passing.** To generate enriched node representations that incorporate both intra- and inter-modal information, we design a crossmodal message passing module guided by the binary attention matrix $\tilde{\alpha}$. This module enables RNA nodes aggregate regulatory cues from attended ATAC nodes, while ATAC nodes receive feedback from their associated RNA targets via the transposed attention map. For each RNA node $r$ and ATAC node $a$, we aggregate information from their relevant nodes:

$$X_{\text{RNA}}^{\text{cross}}[r] = \sum_{a \in \mathcal{V}_{\text{ATAC}}} \tilde{\alpha}_{r,a} \cdot V_{\text{ATAC}}[a], \quad X_{\text{ATAC}}^{\text{cross}}[a] = \sum_{r \in \mathcal{V}_{\text{RNA}}} \tilde{\alpha}_{r,a} \cdot V_{\text{RNA}}[r], \tag{6}$$

To preserve node-specific intrinsic semantics, we incorporate a modality-specific self-update module:

$$\hat{X}_{\text{RNA}}[r] = \texttt{MLP}_{\text{RNA}}(X_{\text{RNA}}[r]) + X_{\text{RNA}}^{\text{cross}}[r], \quad \hat{X}_{\text{ATAC}}[a] = \texttt{MLP}_{\text{ATAC}}(X_{\text{ATAC}}[a]) + X_{\text{ATAC}}^{\text{cross}}[a] \tag{7}$$

where $\texttt{MLP}(\cdot)$ denotes a lightweight feedforward network that captures intra-modal patterns. The result is a set of updated node embeddings containing both intra-modal and inter-modal knowledge.

**Model Training.** We adopt a graph-level binary classification objective, where the label $y \in \{0, 1\}$ indicates whether the RNA and ATAC graphs originate from the same cell. To obtain a compact graph-level representation, we apply average pooling over the final-layer embeddings of the RNA and ATAC nodes, followed by a concatenation and a multi-layer prediction head:

$$\hat{y} = \texttt{Predictor}\left(\texttt{AvgPool}\left(\hat{X}_{\text{RNA}}\right) \| \texttt{AvgPool}\left(\hat{X}_{\text{ATAC}}\right)\right), \tag{8}$$

where $\hat{X}_{\text{RNA}}$ and $\hat{X}_{\text{ATAC}}$ denote the updated embeddings for all expressed RNA and ATAC nodes after message passing in Eq.7, $\texttt{AvgPool}$ is the average pooling operator over rows, $\|$ denotes vector concatenation, and $\texttt{Predictor}$ is a 4-layer feedforward network with ReLU activations.

The model is trained end-to-end by minimizing the binary cross-entropy loss on RNA-ATAC pairs:

$$\mathcal{L} = -y \log \hat{y} - (1 - y) \log (1 - \hat{y}), \tag{9}$$

using the Adam optimizer [23]. This objective encourages the model to learn crossmodal regulatory patterns that are predictive of cell identity alignment. The complexity analysis of Bi$^2$Former is shown in Appendix D, and the limitations are analyzed in Appendix 7.

## 5 Experiments

In this section, we first introduce the datasets and the baselines in Section 5.1. Then we conduct extensive experiments to address the following research questions: **RQ1**: How well does our model perform on the crossmodal matching task? **RQ2**: How robust is our model under limited paired training data? **RQ3**: How effective is the proposed method on the transfer capability across unseen cell types? **RQ4**: What is the contribution of each core component of our model to overall performance? **RQ5**: How do different hyperparameter settings affect model performance? **RQ6**: How can we use our framework for biological interpretation and discovery?

### 5.1 Experimental Setup

**Datasets.** To ensure the reliability and comparability of our evaluation, we conduct experiments on five widely-used benchmark datasets for single-cell omics: ISSAAC-seq [50], 10× Multiome PBMC [1], SHARE-seq [28], SNARE-seq [8], and 10× genomics Multiome. We construct our graph corpus following the description in Section 3. Details of the original datasets and the constructed graphs are in Appendix B.

**Baselines.** We compare Bi$^2$Former against two categories of baselines: (i) VAE-based models, including MultiVI [3], CLUE [35], Cobolt [14], GLUE [6], scMoGNN [47] and scMaui [22], which are originally designed for joint embedding or modality reconstruction. We adapt these models by appending classification heads for the crossmodal matching task. (ii) Methods based on our constructed graph (*i.e.*, ABG) corpus, including a simple MLP, as well as advanced Graph Neural Networks (GNNs) such as GCNII [7], GraphSAGE [16], and Graph Transformer (GT) [11], serving as strong baselines.

Table 1: Crossmodal matching results across various datasets. We report both accuracy and ROC-AUC as evaluation metrics. Boldface indicates the best performance.

| Dataset | ISSAAC-seq | | 10×PBMC | | SHARE-seq | | SNARE-seq | | 10×Multiome | | Avg. | |
| Metric | ACC | ROC-AUC | ACC | ROC-AUC | ACC | ROC-AUC | ACC | ROC-AUC | ACC | ROC-AUC | ACC | AUC |
|---|---|---|---|---|---|---|---|---|---|---|---|---|
| MultiVI | 66.21 ± 1.46 | 69.32 ± 1.07 | 60.93 ± 2.83 | 63.85 ± 1.96 | 64.42 ± 2.19 | 68.87 ± 1.03 | 56.76 ± 1.94 | 61.12 ± 1.18 | 69.35 ± 1.21 | 72.64 ± 1.36 | 63.53 | 67.16 |
| CLUE | 71.28 ± 1.24 | 75.01 ± 0.98 | 68.73 ± 1.67 | 72.26 ± 0.94 | 63.21 ± 2.08 | 67.96 ± 1.19 | 59.32 ± 1.59 | 63.17 ± 0.93 | 73.72 ± 0.97 | 76.92 ± 1.17 | 67.25 | 71.06 |
| Cobolt | 69.21 ± 2.51 | 73.69 ± 1.72 | 61.65 ± 3.05 | 66.74 ± 1.87 | 58.67 ± 3.14 | 61.74 ± 1.62 | 57.46 ± 2.03 | 60.91 ± 1.39 | 71.16 ± 1.44 | 74.15 ± 1.61 | 63.63 | 67.45 |
| GLUE | 74.28 ± 0.91 | 77.40 ± 0.92 | 72.51 ± 1.02 | 79.68 ± 0.71 | 66.89 ± 1.43 | 73.14 ± 1.17 | 64.47 ± 1.22 | 68.28 ± 1.21 | 76.93 ± 0.82 | 80.98 ± 0.92 | 71.01 | 75.90 |
| scMoGNN | 73.72 ± 0.96 | 78.58 ± 0.89 | 72.41 ± 1.37 | 80.76 ± 0.83 | 69.84 ± 1.81 | 74.39 ± 0.94 | 69.03 ± 1.22 | 72.32 ± 0.97 | 75.49 ± 1.31 | 80.04 ± 1.01 | 72.10 | 77.22 |
| scMaui | 71.64 ± 0.97 | 76.19 ± 0.83 | 63.19 ± 2.74 | 67.42 ± 1.52 | 65.93 ± 1.78 | 69.15 ± 0.96 | 58.42 ± 1.65 | 63.14 ± 0.95 | 75.07 ± 0.75 | 78.81 ± 1.13 | 66.85 | 70.94 |
| MLP | 67.39 ± 1.18 | 71.04 ± 0.79 | 62.25 ± 3.74 | 55.87 ± 2.06 | 58.97 ± 0.74 | 62.52 ± 0.57 | 54.74 ± 1.26 | 59.72 ± 1.01 | 70.44 ± 2.07 | 72.62 ± 1.98 | 62.76 | 64.35 |
| GCNII | 72.64 ± 1.29 | 77.32 ± 0.63 | 73.64 ± 0.98 | 79.60 ± 0.54 | 69.49 ± 1.13 | 74.01 ± 0.65 | 62.71 ± 1.07 | 67.93 ± 0.59 | 76.28 ± 1.13 | 81.06 ± 0.76 | 70.95 | 75.98 |
| GraphSAGE | 76.98 ± 0.61 | 82.37 ± 0.35 | 76.92 ± 1.32 | 81.52 ± 0.63 | 67.56 ± 1.24 | 70.93 ± 0.72 | 66.53 ± 0.83 | 70.56 ± 0.57 | 81.94 ± 1.89 | 85.79 ± 1.44 | 73.99 | 78.23 |
| GT | 73.42 ± 0.52 | 80.93 ± 0.31 | 78.04 ± 0.79 | 82.04 ± 0.47 | 72.17 ± 0.53 | 78.34 ± 0.36 | 68.74 ± 0.69 | 73.89 ± 0.42 | 80.12 ± 0.96 | 85.71 ± 0.61 | 74.50 | 80.18 |
| Bi$^2$Former | **84.40 ± 0.48** | **89.24 ± 0.31** | **88.74 ± 0.36** | **92.37 ± 0.16** | **79.84 ± 0.29** | **84.96 ± 0.18** | **73.56 ± 0.37** | **77.30 ± 0.21** | **90.41 ± 0.24** | **93.41 ± 0.30** | **83.39** | **87.46** |

**Other settings.** We report experimental results using hyperparameter settings detailed in Appendix B.4, selecting those that achieve the highest validation performance. While our hyperparameter grids may not always be optimal, they cover a broad range to ensure each model is adequately evaluated on every dataset. Each experiment is repeated with 10 different random seeds, and we report the mean and standard deviation across these runs.

## 5.2 Modal Matching (RQ1)

We evaluate the performance of Bi$^2$Former on the task of crossmodal matching, where the goal is to determine whether a pair of ATAC and RNA sequences originates from the same cell. Table 1 summarizes the results across four benchmark datasets.

First, VAE-based models (*e.g.*, MultiVI, CLUE, Cobolt, GLUE) perform poorly due to their reliance on sparse expression vectors and underutilization of biological attributes. Modeling both expressed and unexpressed elements introduces noise and weakens the meaningful regulatory signals. Second, MLP trained on our constructed ABG corpus benefits from denoised inputs and biological attributes, achieving modest results. However, its lack of structural and interaction-aware modeling limits its performance. Third, GNN models (*i.e.*, GCNII, GraphSAGE, GT) further improve performance by leveraging structural information. However, their inductive biases are typically locality-driven and lack explicit mechanisms to model biologically meaningful crossmodal interactions.

Bi$^2$Former addresses these limitations by a biologically-driven crossmodal attention mechanism that filters low-confidence signals, incorporates chromosomal priors, and aligns with the biological truth. Furthermore, our crossmodal message passing preserves intra-modal semantics while capturing inter-modal dependencies, enabling Bi$^2$Former to capture fine-grained interaction patterns that general GNNs cannot model explicitly. As a result, Bi$^2$Former consistently outperforms baselines, surpassing the strongest VAE baseline by an average of 11.3% and the strongest ABG-based baseline by 8.9% in accuracy, with the largest improvement of 16.2% and 10.7% on 10×PBMC. These gains underscore the strength of our biologically grounded framework and its generalizability across diverse datasets.

## 5.3 Robustness under Sparse Supervision (RQ2)

To evaluate the ability to effectively handle sparsely paired datasets, which is a significant challenge in biological measurement data, we conducted experiments comparing our model with other baselines under different training set sizes. As shown in Figure 3, VAE-based methods experience a notable performance drop when the amount of paired training data is reduced. In particular, when only 20% of the training pairs are available, most baselines lose the ability to effectively distinguish positive from negative samples. By contrast, Bi$^2$Former maintains strong performance even under such low-resource settings. This robustness arises from its biological attribute-aware design and its objective that explicitly captures fine-grained crossmodal interaction patterns, enabling efficient utilization of limited training data.

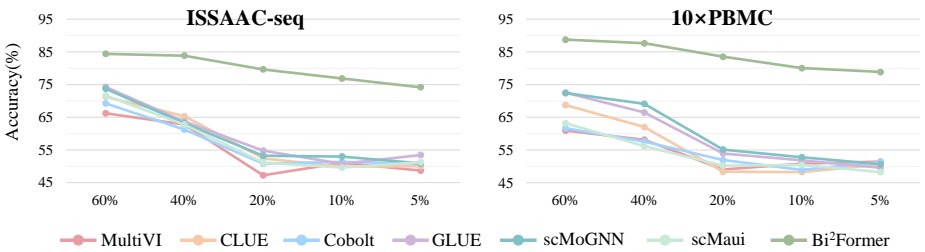

Figure 3: Results for crossmodal matching task with different training sizes.

Table 2: Across-cell-types prediction results for crossmodal matching across various datasets. We report accuracy as evaluation metric. Boldface indicates the best performance.

| Dataset | ISSAAC-seq | 10×PBMC | SHARE-seq | SNARE-seq |
|---|---|---|---|---|
| MultiVI | 47.32 ± 1.39 | 54.72 ± 1.68 | 52.81 ± 1.51 | 51.83 ± 2.15 |
| CLUE | 61.28 ± 1.16 | 61.41 ± 1.31 | 57.31 ± 1.32 | 56.45 ± 1.82 |
| Cobolt | 57.84 ± 2.01 | 58.67 ± 1.96 | 55.27 ± 2.47 | 48.63 ± 2.51 |
| scMaui | 62.43 ± 1.84 | 58.54 ± 1.79 | 54.64 ± 2.05 | 52.49 ± 1.74 |
| MLP | 65.72 ± 1.13 | 61.84 ± 1.69 | 58.74 ± 1.12 | 53.67 ± 1.33 |
| GCNII | 72.18 ± 1.29 | 72.52 ± 0.78 | 67.37 ± 0.96 | 61.38 ± 1.27 |
| GraphSAGE | 74.15 ± 0.93 | 76.31 ± 0.65 | 67.02 ± 1.31 | 64.56 ± 0.92 |
| GT | 71.92 ± 0.82 | 76.43 ± 0.81 | 69.94 ± 0.95 | 67.85 ± 0.81 |
| Bi²Former | **82.74 ± 0.74** | **84.96 ± 0.49** | **78.07 ± 0.61** | **71.28 ± 0.32** |

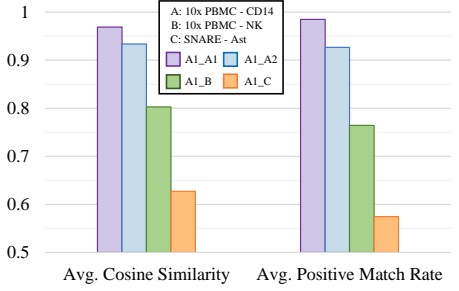

Figure 4: Cross-model similarity of attention matrices. The metrics are computed between attention matrices of the same held-out cells from models with different train corpus.

## 5.4 Cross-Cell-Type Generalization (RQ3)

**Transfer Capability across Unseen Cell Types.** To assess the generalization capability of our method, we evaluate the performance of Bi²Former under a cross-cell-type setting. Specifically, we split each dataset into training and test sets with disjoint cell types in a 1:1 ratio (See details in Appendix B.3). This setup ensures that the model is evaluated on completely unseen biological categories. As shown in Table 2, ABG-based methods significantly outperform traditional VAE-based baselines, emphasizing the benefits of denoised signals and biological attributes for improved generalization. Moreover, Bi²Former consistently achieves the best performance across all settings, surpassing the strongest ABG-based baseline by an average of 7.7%, highlighting the benefits of the biologically-driven design and its strong ability to generalize across cell types.

**Analysis of Learned Attention Matrices across Cell Types.** Beyond prediction accuracy, we further investigate whether the attention learned by Bi²Former captures transferable biological regulatory mechanisms across different cell types. Specifically, we train separate models on corpus of different cell-type. Then we compare the attention matrices generated for the same test cells among these models. As shown in Figure 4, attention matrices generated by models trained on different groups of the same cell type (*i.e.*, A1 and A2) remain highly consistent, with average cosine similarity of 0.93. Moreover, models trained on A1 (one type of blood mononuclear cells from PBMC) and B (another type of blood mononuclear cells from PBMC) yield relatively similar regulatory patterns (0.80), whereas those trained on A1 and C (one type of neuronal tissue cells from SNARE) show much lower similarity (0.63). These results suggest that Bi²Former captures cell-type-specific regulatory interaction patterns that generalize more effectively across biologically similar populations.

## 5.5 Ablation Study (RQ4)

To understand the contribution of each key component in Bi²Former, we conduct an ablation study by selectively masking node ID embeddings (*i.e.*, $ID(v)$ in Eq.(2)), biological attributes (*i.e.*, $Expr(v)$ and $BioAttr(v)$ in Eq.(2)), biological pruning (BP) in the crossmodal attention module, and the edge structure informed by prior biological knowledge in the graph corpus.

Results are shown in Table 3. Interestingly, removing ID embeddings leads to a more significant performance drop than removing biological attributes. This suggests that with sufficient training

Table 3: Ablation study of masking different components in Bi$^2$Former.

| Methods | ISSAAC-seq | 10×PBMC | SHARE-seq | SNARE-seq | Avg.$\Delta$ |
|---|---|---|---|---|---|
| Bi$^2$Former | 84.40 ± 0.48 | 88.74 ± 0.36 | 79.84 ± 0.29 | 73.56 ± 0.37 | - |
| w/o ID | 80.64 ± 1.51 | 83.06 ± 2.04 | 76.75 ± 1.87 | 70.97 ± 1.01 | ↓ 3.78 |
| w/o Attribute | 81.47 ± 0.49 | 84.32 ± 0.38 | 77.32 ± 0.31 | 71.74 ± 0.47 | ↓ 2.93 |
| w/o BP | 83.87 ± 0.64 | 85.79 ± 0.46 | 77.97 ± 0.31 | 73.21 ± 0.36 | ↓ 1.43 |
| w/o Edge | 78.89 ± 0.57 | 83.68 ± 0.49 | 76.72 ± 0.28 | 70.15 ± 0.39 | ↓ 4.28 |
| w/o Attribute, BP, and Edge | 72.39 ± 1.12 | 70.86 ± 1.74 | 69.78 ± 2.14 | 67.45 ± 1.13 | ↓ 11.52 |

data, the model is able to effectively learn the interaction patterns between nodes through repeated exposure—leveraging latent co-occurrence signals across graphs, highlighting the model's ability to infer relationships in a data-driven manner under fully supervised conditions.

We next remove the biological pruning, *i.e.*, the sigmoid-threshold activation and top-$k$ mask, which is designed to suppress low-confidence signals and reflect the biological constraint that genes are typically regulated by a limited number of ATAC peaks. This results in a modest performance drop, indicating that our attention sparsification pruning is able to align model with biological priors.

Furthermore, we evaluate the impact of removing the edges by replacing the prior-based adjacency matrix with a fully connected bipartite graph between expressed ATAC and RNA nodes. This design removes biological priors and allows unrestricted attention computation across all node pairs. The performance degrades but remains competitive compared to the VAE-based baselines. Overall, these confirm two points: (i) our attention mechanism is capable of learning useful interactions even under noisy topologies, and (ii) biologically grounded edge priors serve as an effective inductive bias, guiding the model toward more interpretable and accurate regulatory patterns.

Finally, we remove the attributes, biological pruning, and edges, retaining only the expressed nodes. This modification results in the model only filtering out unexpressed nodes compared to traditional VAE-based models. We observed a significant performance drop, but the model still outperformed the strongest VAE-based baseline. This further emphasizes the effectiveness of filtering out noise and focusing solely on the co-occurrence patterns on expressed nodes.

## 5.6 Hyperparameter Study (RQ5)

We investigate the influence of key hyperparameters in Bi$^2$Former, focusing on the threshold strategy and the top-$k$ selection strategy within the Biologically-driven Crossmodal Attention module.

**Threshold $\tau$.** The threshold parameter $\tau$ is introduced to filter out low-confidence attention signals. This design is motivated by our early observations that, during the initial training stages, under-trained nodes tend to produce uniformly distributed attention weights, which introduces considerable noise and deviates from biologically meaningful regulatory patterns. To address this, we first set the threshold $\tau = 0.5$, which effectively suppresses these uniformly noisy distributions. We then gradually increased the threshold to assess its impact on model performance. As shown in Figure 5, the optimal threshold varies slightly across different datasets, likely reflecting biological differences in RNA–ATAC interaction sparsity across distinct cell types. Higher thresholds enforce stricter gating, allowing the most confident and specific regulatory links to be preserved.

**Top-$k$.** The top-$k$ sparsification strategy following thresholding is introduced to further align with the biological assumption that each gene is regulated by a limited number of cis-regulatory elements. We experiment with $k \in \{5, 10, 15, 20\}$, and the results shown in Figure 5 indicate that $k = 10$ performs the best across most datasets. This finding supports the notion that a small number of ATAC regions contribute significantly to RNA regulation, aligning well with known biological priors.

## 5.7 Biological Interpretation and Discovery (RQ6)

**Cell-level.** In Figure 4, we evaluate the plausibility of the learned attention matrix from a computational perspective. To further interpret Bi$^2$Former from a biological standpoint, we leverage the learned RNA–ATAC attention matrix $\tilde{\alpha}$ as a proxy for regulatory interactions. At the cell-level, $\tilde{\alpha}$ reveals how accessible ATAC potentially regulate gene expression (RNA), enabling fine-grained

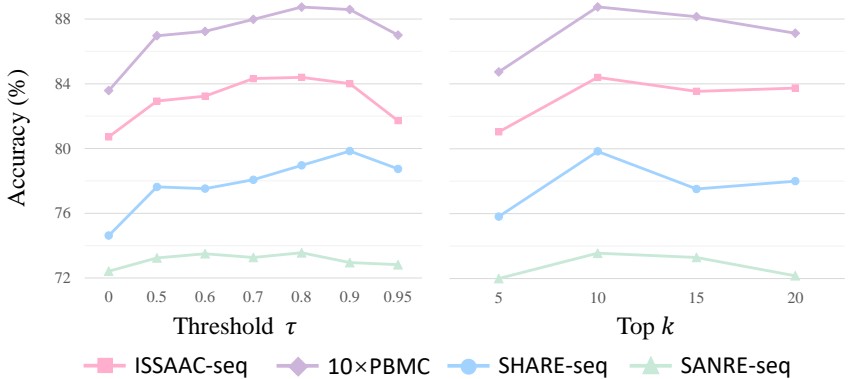

Figure 5: Experiment results of the hyperparameter of Bi$^2$Former.

cell-specific interpretation. As shown in Appendix C.1, each RNA is regulated by a limited number of ATAC peaks, consistent with known biological principles of gene regulation.

**Cell-type-level.** By aggregating attention matrices across cells of the same type, we obtain population-level regulatory maps that reflect cell-type-specific transcriptional programs. These insights facilitate the comparative analysis of regulatory patterns across cell types. Moreover, to further validate the biological relevance of our model, we compare the cumulative attention scores against experimentally derived TF binding scores [21], which reflect the actual activation strength of ATAC peaks in each cell type and serve as a proxy for the ground truth. The results show that the cumulative attention scores for CD4 cells exhibit strong agreement with CD4-specific TF binding signals, indicating that our model successfully identifies biologically meaningful regulatory relationships. Details are provided in Appendix C.2.

## 6   Conclusion

In this work, we present a novel framework that formulates single-cell crossmodal matching as an interaction learning problem via graph classification task on Attributed Bipartite Graphs (ABGs). Our study introduces an interpretable ABG-based approach to single-cell crossmodal analysis, paving the way for more structured and insightful crossmodal learning in biology. This perspective allows for explicit modeling of interaction while leveraging both observed data and node-level features. To model the regulatory interactions on these graphs, we propose Bi$^2$Former, a biologically-driven bipartite graph transformer that learns interpretable attention over potential ATAC–RNA regulatory pairs, explicitly modeling crossmodal interactions. Extensive experiments across diverse datasets show that our model achieves state-of-the-art performance in crossmodal matching, generalizes well to unseen cell types, and uncovers biologically meaningful regulatory interactions. Our study introduces an interpretable ABG-based approach to single-cell crossmodal analysis, paving the way for more structured and insightful crossmodal learning in biology.

## 7   Limitations

While Bi$^2$Former achieves strong performance and interpretability, it does not currently incorporate rich edge attributes, which can play a crucial role in capturing fine-grained interactions in graph-based tasks [19]. Modeling such edge-level information requires the integration of more detailed biological priors and suitable encoding strategies. In future work, we plan to extend our framework by introducing biologically informed edge attributes to fully exploit their representational power.

## Acknowledgements

This work is supported by Guangdong Basic and Applied Basic Research Foundation project 2025A1515010304, Guangzhou Science and Technology Planning Project 2025A03J4491, National Key Research and Development Program of China (2024YFA1307703 to H.L.).

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

Table 4: Statistics of the original datasets.

| Methods | #RNA | #ATAC | #Cell (Positive Sample) | #Cell Types |
|---------|------|-------|-------------------------|-------------|
| ISSAAC-seq | 32,208 | 169,180 | 10,361 | 23 |
| 10×PBMC | 29,095 | 107,194 | 9,631 | 19 |
| SHARE-seq | 21,478 | 340,341 | 32,231 | 22 |
| SNARE-seq | 28,930 | 241,757 | 9,190 | 22 |
| 10×Multiome | 13,431 | 116,490 | 69,249 | 22 |

# A  Related Works of Graph Neural Networks and Graph Transformers

Graph neural networks (GNNs) [25, 7, 39, 16, 44] propagate and aggregate neighborhood information through message passing, making them well-suited for biological fields [5, 26, 52, 51]. In the heterogeneous graph setting [49, 20, 42], such mechanisms allow flexible information fusion across node types and modalities, which aligns with our design of RNA-ATAC bipartite graphs.

Recently, graph transformers [11, 48, 32, 54] have gained popularity due to their global receptive fields and capacity to model complex dependencies. Attention not only captures relations but also enhances interpretability by quantifying the importance of each interaction [45, 34, 53].

Building on the ABG framework, we design a biologically-driven crossmodal graph transformer tailored to the single-cell omics context. By incorporating biological priors and biological pruning, our model learns fine-grained regulatory patterns between ATAC and RNA. The attention module is carefully designed to highlight interpretable crossmodal signals, enabling us to uncover meaningful ATAC-RNA interactions patterns at both the cell-level and cell-type-level. Unlike generic graph attention methods, our approach grounds the attention weights in biological relevance, offering interpretability beyond performance.

# B  Experimental Details

## B.1  Details of the Original Datasets

We evaluate $Bi^2$Former on four widely-used single-cell omics datasets that provide paired scRNA-seq and scATAC-seq profiles from the same cells. Summary statistics are provided in Table 4.

**ISSAAC-seq.**  ISSAAC-seq [50] is a large-scale human multi-omics dataset that jointly profiles chromatin accessibility and gene expression at single-cell resolution. It contains over 10,000 cells spanning 23 immune and epithelial cell types, making it suitable for evaluating both matching performance and generalization across diverse cell identities.

**10× PBMC.**  The 10x Multiome PBMC dataset [1] includes peripheral blood mononuclear cells from healthy donors, providing paired ATAC and RNA modalities with moderate sparsity and cell-type diversity. It is a benchmark dataset in many multimodal learning studies.

**SHARE-seq.**  SHARE-seq [28] is one of the largest publicly available paired multi-omics datasets, capturing chromatin accessibility and gene expression across over 30,000 cells. Due to its large peak count and sparse feature distributions, it is particularly challenging for cross-modal modeling.

**SNARE-seq.**  SNARE-seq [8] enables simultaneous profiling of RNA and chromatin accessibility, particularly focused on neural tissues. Despite a moderate sample size, its high ATAC dimensionality and tissue-specific regulatory features make it useful for evaluating biological interpretability.

**10× Multiome.**  The 10x Genomics Multiome dataset [27] originates from the 2021 NeurIPS Open Problems in Single-Cell Analysis competition. The dataset comprises several thousand cells spanning major immune cell types, with moderate feature sparsity and well-balanced cell-type representation. These properties make it a widely used benchmark for evaluating crossmodal matching, joint embedding, and modality-prediction methods under realistic paired-data conditions.

Table 5: Statistics of our ABG datasets.

| Methods | #Graphs | Avg.#RNA Nodes | Avg.#ATAC Nodes | Avg.#Edges | Avg.Sparsity | Split(%) |
|---|---|---|---|---|---|---|
| ISSAAC-seq | 20,722 | 1,843 | 7,578 | 851,136 | 0.06 | 60/20/20 |
| 10×PBMC | 19,262 | 1,924 | 7,379 | 762,927 | 0.05 | 60/20/20 |
| SHARE-seq | 64,462 | 619 | 3,971 | 157,132 | 0.06 | 60/20/20 |
| SNARE-seq | 18,380 | 937 | 2,452 | 133,146 | 0.05 | 60/20/20 |

## B.2 Details of the ABG Datasets

The summary statistics of our constructed ABG corpus are detailed in Table 5.

As described in Section 3, during graph construction, we encode biological attributes into the features of each RNA and ATAC node by `BioAttr`. Specifically, RNA nodes include attributes: $\{'chrom','means','variances\_norm','strand','highly\_variable'\}$, and ATAC nodes include: $\{'chrom','dna\_sequence'\}$. Categorical and Statistics attributes are processed via one-hot encoding and direct numerical normalization, respectively. The $dna\_sequence$ is truncated to 256 dims and encoded using a sequential encoding scheme.

While matched pairs reflect true biological interaction patterns across modalities, mismatched pairs are crucial for learning a robust decision boundary. They help the model distinguish meaningful alignments from random co-occurrence. This discrimination is particularly important given the high dimensionality and noise in single-cell omics data. To preserve biological diversity and avoid sampling bias, negative samples are drawn proportionally according to the distribution of cell types in the dataset. As summarized in Table 5, we maintain a 1:1 ratio of positive to negative pairs, resulting in a graph dataset that contains twice the number of samples as the original single-cell dataset. Full implementation details are available in our code repository.

## B.3 Details of the Cross-cell-types Setting

To assess the generalization capability of our method, we evaluate the performance of Bi$^2$Former under a cross-cell-type setting. We split each dataset into training and test sets with disjoint cell types in a 1:1 ratio. To ensure a balanced partition, we sort all cell types by their number of cells and assign those at odd and even indices to the training and test sets, respectively. Specifically:

**ISSAAC-seq.** Training cell types: {"R3 Ex-L5 IT", "R13 In-Drd2", "R8 Ex-L6 IT Bmp3", "R16 In-Sst", "R10 Ex-L6b", "R21 Oligo"}; Test cell types: {"R7 Ex-L6 CT", "R12 Misc", "R6 Ex-L5-PT", "R20 OPC", "R9 Ex-L6 IT Oprk1", "R22 VLMC"}.

**10× PBMC.** Training cell types: {"CD14 Mono", "CD8 Naive", "CD16 Mono", "CD8 TEM_1", "Intermediate B", "CD4 TEM", "Treg", "MAIT", "pDC", "Plasma"}; Test cell types: {"CD4 Naive", "CD4 TCM", "NK", "CD8 TEM_2", "Memory B", "cDC", "gdT", "Naive B", "HSPC"}.

**SHARE-seq.** Training cell types: {"Basal", "TAC-1", "CD16 Mono", "alow CD34+ bulge", "Hair Shaft-Cuticle/Cortex", "ORS", "Medulla", "Dermal Papilla", "IRS", "Dermal Sheath", "Macrophage DC", "Sebaceous Gland"}; Test cell types: {"Infundibulum", "Spinous", "ahigh CD34+ bulge", "Dermal Fibroblast", "TAC-2", "Endothelial", "Isthmus", "K6+ Bulge/Companion Layer", "Granular", "Melanocyte", "Schwann Cell"}.

**SNARE-seq.** Training cell types: {"E2Rasgrf2", "E6Tle4", "E5Galnt14", "Ast", "InP", "E3Rmst", "InS", "InV", "OPC", "Mic", "Endo"}; Test cell types: {"E3Rorb", "E4Il1rapl2", "E4Thsd7a", "E5Parm1", "OliM", "E5Sulf1", "Clau", "InN", "OliI", "Peri"}.

## B.4 Hyperparameters

Experimental results are reported on the hyperparameter settings below, where we choose the settings that achieve the highest performance on the validation set. We choose hyperparameter grids that do

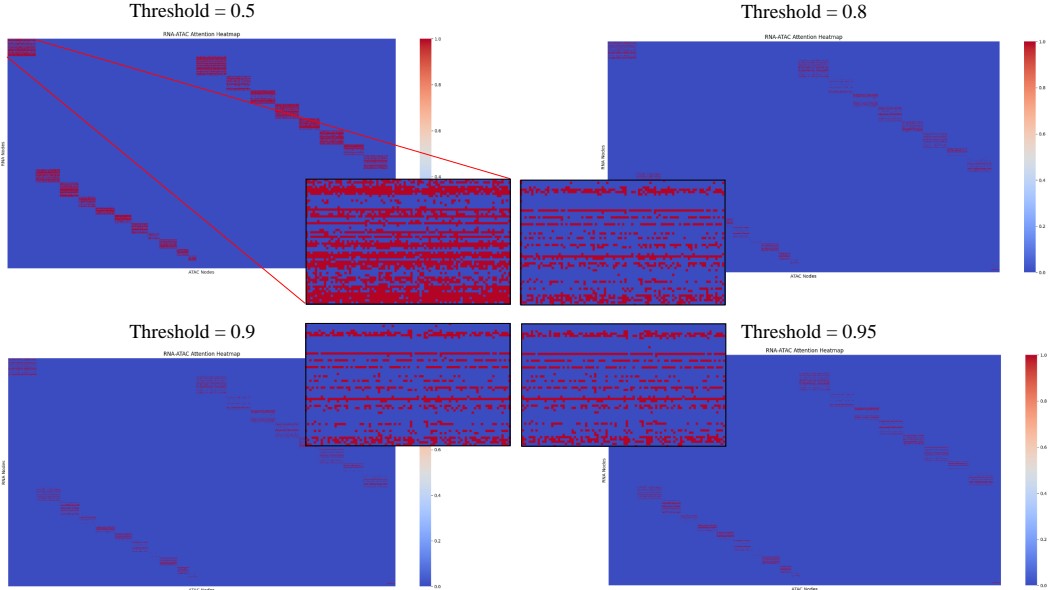

Figure 6: Case study of RNA–ATAC attention matrices for a representative single cell under different threshold values $\tau$. Higher thresholds remove noisy interactions and highlight confident regulatory links.

not necessarily give optimal performance, but hopefully cover enough regimes so that each model is reasonably evaluated on each dataset.

- learning_rate $\in \{1e-3, 5e-4, 1e-4, 5e-5, 1e-5\}$

- weight_decay $\in \{1e-4, 5e-5, 1e-5, 5e-6, 1e-6\}$

- dropout $\in \{0, 0.1, 0.3, 0.5, 0.8\}$

For Bi2Former,

- ID embedding dims $\in \{64, 128, 256, 512\}$

- hidden dims $\in \{64, 128, 256, 512\}$

- layer_num $\in \{1, 2\}$

## C  Case of Biological Interpretation and Discovery

### C.1  Cell-Level Regulatory Interaction.

To further demonstrate the interpretability of $\text{Bi}^2\text{Former}$ and its capacity to reveal interaction between ATAC and RNA, we conduct a case study visualizing the learned RNA–ATAC attention matrix $\tilde{\alpha}$ under varying threshold settings for a representative single cell. As shown in Figure 6, increasing the threshold $\tau$ progressively suppresses low-confidence signals, resulting in a sparser attention matrix and more specific and meaningful regulatory signals.

However, we observe that without further constraints, some RNA nodes may either lack any activated ATAC connections or remain dense connected—both of which deviate from biological priors. To address this, we introduce a top-$k$ constraint following thresholding. After training with top-$k$ regularization, the model not only achieves better performance but also produces more interpretable attention patterns. As illustrated in Figure 7, each RNA is regulated by a limited number of ATAC peaks, consistent with known biological principles of gene regulation.

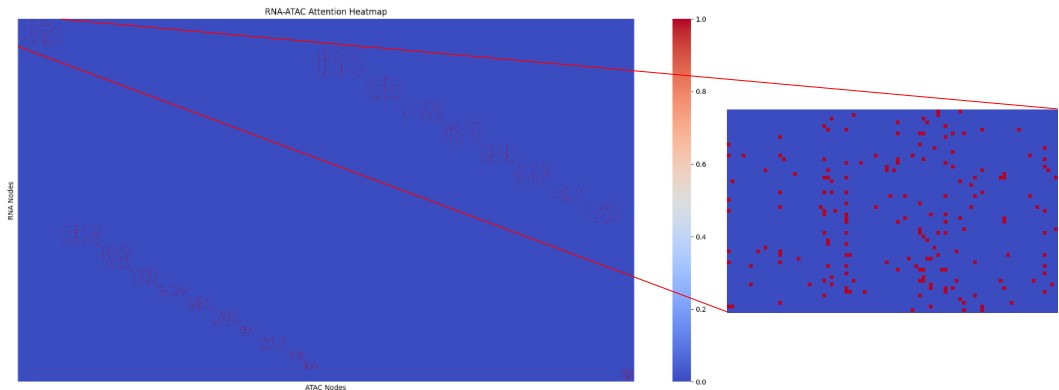

Figure 7: Improved RNA–ATAC interaction map after applying top-$k$ refinement. Each RNA is connected to a limited number of ATAC peaks, aligning with known biological regulation mechanisms.

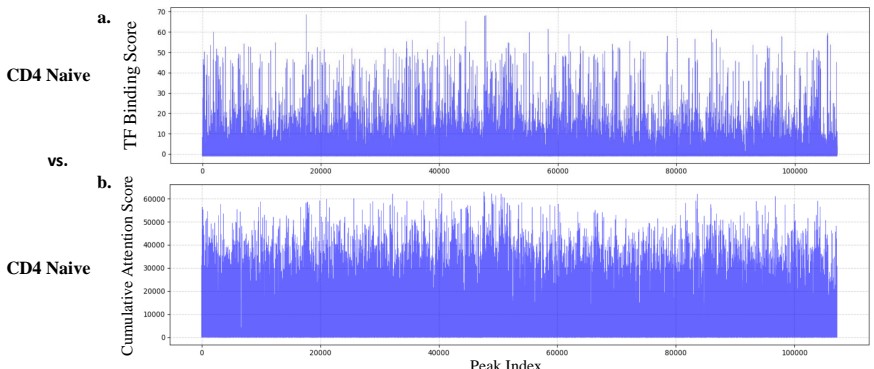

**Cosine Similarity: 0.608   Cosine similarity on ranks: 0.909   Pearson Correlation Coefficient: 0.633   Jaccard Index (TOP10000): 0.548**

Figure 8: Comparison between TF binding scores and cumulative attention scores within the same cell type (CD4). The cumulative attention scores produced by our model align closely with the TF binding intensity in CD4 cells, suggesting that the learned crossmodal interactions accurately capture cell-type-specific regulatory signals.

## C.2   Cell-Type-Level Regulatory Maps.

To investigate how ATAC–RNA regulatory patterns vary across cell types, we aggregate ATAC expression signals across cells of the same type and analyze the resulting attention-informed regulatory maps. Specifically, we compute the cumulative attention score as the regulatory strength of each ATAC peak across different cell types to identify cell-type-specific activation patterns.

As shown in Figure 8.b and Figure 9.b, clear differences emerge between CD4 and CD14 cell populations, revealing divergent regulatory patterns that reflect their distinct biological functions. Moreover, to further validate the biological relevance of our model, we compare the cumulative attention scores against experimentally derived TF binding scores [21], which reflect the actual activation strength of ATAC peaks in each cell type and serve as a proxy for ground truth. In Figure 8, the cumulative attention scores for CD4 cells exhibit strong agreement with CD4-specific TF binding signals, indicating that our model successfully identifies biologically meaningful regulatory relationships. In contrast, Figure 9 shows that CD14 cells exhibit different attention profiles relative to the CD4 ground truth, underscoring cell-type-specific regulatory patterns. These results highlight the capacity of our approach to uncover interpretable and biologically grounded crossmodal interactions at the population level.

Overall, these cell-type-level maps offer a powerful means to dissect cell identity through the strength of ATAC peak signals.

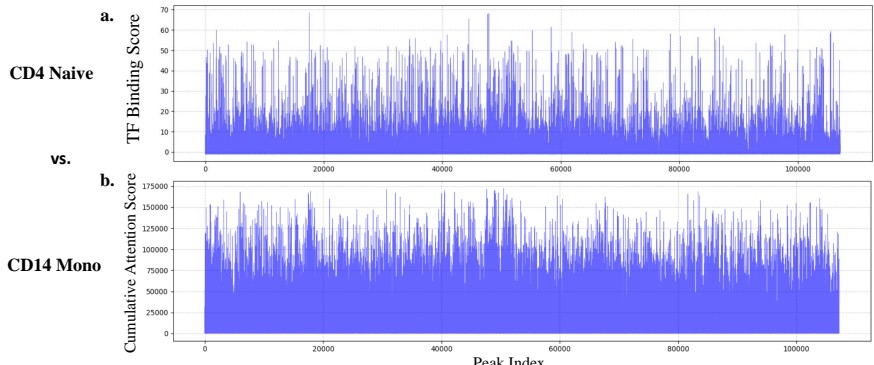

**Cosine Similarity: 0.330   Cosine similarity on ranks: 0.674   Pearson Correlation Coefficient: 0.358   Jaccard Index (TOP10000): 0.302**

Figure 9: Comparison between TF binding scores from CD4 cells and cumulative attention scores in CD14 cells. CD14 cells exhibit distinct peak activation patterns with the CD4 ground truth, highlighting the distinct regulatory patterns across cell types.

# D   Complexity Analysis

In this section, we present the time and space complexity analysis of Bi$^2$Former. For simplicity, we assume that the feature dimension remains unchanged and that the number of model layers is set to 1, and the $N_r$ and $N_a$ denote the number of RNA and ATAC nodes in the graph, *i.e.*, $|\mathcal{V}_{\text{RNA}}|$ and $|\mathcal{V}_{\text{ATAC}}|$. $E$ is the number of edges (*i.e.*, the number of non-zero entries in the adjacency matrix $A$), and $d_h$ is the hidden dimension.

## D.1   Time Complexity

The primary time overhead arises from three components: the Biologically-driven Crossmodal Attention, the Crossmodal Message Passing, and the Predictor. The time complexity of the Crossmodal Attention is $\mathcal{O}(N_r d_h^2 + N_a d_h^2 + E d_h)$. The Crossmodal Message Passing aggregates messages from sparse attention edges and includes self-attentions, with complexity $\mathcal{O}((N_r + N_a)d_h^2 + E d_h)$. Finally, the complexity of the Predictor is a negligible overhead of $\mathcal{O}(d_h^2)$. Thus the total time complexity of our method is $\mathcal{O}((N_r + N_a)d_h^2 + E d_h)$.

Compared with VAE-based methods, which operate on full dense expression matrices with time complexity $\mathcal{O}(N d_h^2)$ (where $N$ is the total number of ATAC and RNA, typically tens of times larger than the number of expressed $N_r + N_a$ per graph), our method is significantly more efficient.

## D.2   Space Complexity

For each graph, we maintain hidden node embeddings, sparse attention matrices, and aggregated messages. Specifically, the feature storage and value projections require $\mathcal{O}((N_r + N_a)d_h)$; and the attention matrix and message buffers take $\mathcal{O}(E d_h)$. Hence, the overall space complexity is $\mathcal{O}((N_r + N_a)d_h + E d_h)$.

Compared with VAE-based methods with space complexity $\mathcal{O}(N d_h)$, our method reduces unnecessary memory usage on unexpressed nodes, leading to significantly improved memory efficiency.

