# OpenReview forum: "Learning Crossmodal Interaction Patterns via Attributed Bipartite Graphs for Single-Cell Omics"
_NeurIPS.cc/2025/Conference — NeurIPS 2025 poster_

### Official Review · Reviewer_7QNJ · 2025-07-02

**Clarity:** 3
**Significance:** 2
**Originality:** 3
**Rating:** 4
**Confidence:** 3

**Summary:**

This paper introduces a method for crossmodal matching in single-cell omics by casting it as a graph classification task on Attributed Bipartite Graphs. The authors model single-cell omics data ( ATAC and RNA) modalities, as ABG with nodes representing individual ATAC and RNA features, incorporating IDs and biological attributes. They propose a transformer architecture to learn interpretable attention over ATAC and RNA pairs, aiming to capture regulatory relationships. Empirical experiments are conducted on several datasets to validate the method performance in cross-modal matching, its generalization to unseen cell types, and its ability to capture biologically meaningful regulatory patterns.

**Questions:**

1. Can the authors discuss the robustness of the method against the problem of unpaired data? Is training on large paired data required to achieve good performance? Can it handle unpaired or semi-paired settings?
2. The ABG and transformer-based architecture might be computationally costly. Can the authors discuss this point?
3. Can the proposed method generalize to other modalities beyond ATAC and RNA, such as methylation data, and what modifications would be needed?
4. How sensitive is the method is to  to the negative sampling strategy as this can be related to robustness agains unpaired data, and does this affect its generalization?

**Ethical Concerns:**

["NO or VERY MINOR ethics concerns only"]

**Final Justification:**

I give a score of 4 and recommend acceptance. The paper presents a well-motivated methodological contribution to the problem of cross-modal matching in single-cell omics, an important challenge in the field. It is clearly written, the limitations of prior work are well identified, and the proposed method is supported by strong experimental results. In particular, the new experiments demonstrating the method's effectiveness with a low number of paired omics samples are convincing. Emphasizing this strength more clearly would further improve the paper.

**Limitations:**

The paper discuss the limitations of their method. I encourage the authors to revise this section, as reviewer feedback could help improve the limitations this section.

**Paper Formatting Concerns:**

No issue noticed.

**Quality:**

3

**Strengths And Weaknesses:**

**Strengths**

* The paper is methodological work studying the problem of cross-modal matching in single-cell omics, which is important for both researchers and practitioners.
* The paper is well written and easy to read. The challenges and limitations of existing literature are clearly stated and the proposed solutions are well motivated.
*The experimental campaign validates the method performance and supports the presented claims.

**Weaknesses**

* The paper assumes access to paired ATAC and RNA data. However, it is well known that these omics require destructive sequencing techniques, so obtaining paired data is challenging. The paper does not discuss this limitation, which may affect the utility of the method.
* Only two modalities are considered. While ATAC and RNA are popular, it would be interesting to see extensions of the method to other omics data.

---

> ### Author Rebuttal · Authors · 2025-07-31
>
> Dear Reviewer 7QNJ,
>
> Thanks for your constructive and encouraging comments. Below we address your concerns.
>
> ***
>
> ## **W1 & Q1.1 & Q1.2 About the Unpaired Data**
> Thank you for your feedback and valuable suggestions.
> While it is true that paired scRNA-seq and scATAC-seq data (by sequencing techniques) are required during training (which is necessary during training for almost all crossmodal integration models), our model is designed to be **highly data-efficient and generalizable**. By incorporating biological attributes as node representations and explicitly modeling crossmodal interaction patterns, our method requires fewer paired samples than previous approaches while achieving superior performance and strong generalization. To further address your concerns, we conducted experiments comparing our model with other baselines under different training set sizes on the ISSAAC-seq dataset:
>
> > Table I: Results for crossmodal matching task with different training sizes on ISSAAC-seq dataset.
>
> Training Set Scale|60%|40%|20%|10%
> :-|:-:|:-:|:-:|:-:
> MultiVI|66.21|62.79|47.24|51.07
> CLUE|71.28|65.32|52.41|49.86
> scMoGNN|73.72|63.15|53.22|51.98
> Bi2Former(Ours)|84.40|83.84|79.63|72.85
>
> As shown in the table, VAE-based methods experience a notable performance drop when the amount of paired training data is reduced. In contrast, Bi2Former maintains strong performance even in low-resource settings, thanks to its biological attribute-aware design and its ability to capture fine-grained interaction patterns. We will expand the relevant discussion in the Limitation section of the revised version to clarify this point.
>
> ## **Q1.3 Can it Handle Unpaired or Semi-paired Settings?**
> Our model is naturally compatible with both semi-paired and unpaired scenarios, and we have empirically validated its robustness and generalization in these settings.
>
> Our cross-cell-type generalization experiments (Section 5.3, Table 2) effectively simulate a **semi-paired setting**. During training, we utilize paired samples (along with corresponding negative samples) from only a subset of cell types. At test time, the model is evaluated on unseen cell types, whose data have no pairing information available in the training set.
>
> In addition, we evaluate our model under a zero-shot cross-dataset setting, which is essentially an **unpaired setting**. Specifically, we pretrained the model on source datasets and evaluated it directly on different target datasets without any fine-tuning. This setup simulates a realistic deployment scenario where no pairing information is available during inference. The results are summarized below:
>
> > Table C: Zero-shot inference results for crossmodal matching task with training across various source dataset(s). The target dataset is ISSAAC-seq.
>
> Source dataset(s)|10xMultiome-PBMC|10xMultiome-PBMC+SHARE-seq|10xMultiome-PBMC+SHARE-seq+SNARE-seq
> :-|:-:|:-:|:-:
> MultiVI|47.82|48.19|51.24
> CLUE|51.07|49.67|52.93
> scMoGNN|54.74|61.02|58.89
> Bi2Former(Ours)|71.14|79.16|80.03
>
> As shown, our model outperforms all baselines significantly even when trained on a single source dataset (71.14% with one source vs. 58.89% with three sources). Moreover, as we incorporate more training data from additional source datasets, performance continues to improve, highlighting our model’s strong generalization capability. This improvement stems from the model’s ability to leverage biological attributes and learn biologically meaningful crossmodal interaction patterns, rather than relying solely on direct paired examples.
>
> These results strongly support the applicability of our method to both semi-paired and fully unpaired crossmodal settings.
>
> ***
>
> ## **Q2 About the Computational Cost**
> As detailed in Appendix D (Line 564-579), we provide a **theoretical analysis** of the time and space complexity of our model. Although Bi2Former computes attention matrices, the computational cost is significantly alleviated by two key strategies:
> - **Sparse Edge**: Attention is computed only on biologically meaningful edges, rather than across all node pairs.
> - **Node Reduction** via Expressed Features: The ABG include only the expressed RNA and ATAC features as nodes, which substantially reduces graph size and computation.
>
> Thus, Bi2Former is computationally **more efficient than traditional VAE-based methods**. Specifically, the computational complexity of Bi²Former is $O((N_a + N_r)d^2 + Ed)$, which is more favorable compared to $O(Nd^2)$ for typical VAE-based methods (where $N$ is the total number of ATAC and RNA, typically tens of times larger than the number of expressed $N_r + N_a$ per graph).
>
> To further address your concern, we also empirically report the training time per epoch (s/epoch) on the ISSAAC-seq and SHARE-seq datasets:
>
> ||Time Complexity|Time(ISSAAC)|ACC(ISSAAC)|Time(SHARE)|ACC(SHARE)
> :-|-|-|-|-|-
> MultiVI|$O(Nd^2)$|218.7|66.21|732.1| 64.42
> CLUE|$O(Nd^2)$|305.1|71.28|1031.9|63.21
> Bi2Former(Ours)| $O((N_a + N_r)d^2 + Ed)$|**121.2**|**84.40**|**365.3**|**79.84**
>
> Analysis: Bi2Former has the outstanding performance while maintaining competitive running times.
>
> ***
>
> ## **W2 & Q3 Generalization to Other Modalities**
> ### 1). Two modalities
> Our framework is **not limited to scRNA-seq and scATAC-seq**. The Bi²Former architecture and its biological-driven attention mechanism are general and can be directly applied to other modality pairs (e.g., RNA–Protein, ATAC–Methylation) for tasks such as matching, generation, or interpretability with only minor modifications.
> Specifically, the only modifications needed are in the biological attributes of nodes and the edge construction strategy, depending on the modalities involved. The core model architecture remains unchanged, showcasing the flexibility and extensibility of our approach. To further address your concern, we also empirically report the RNA-Protein matching results:
>
> > Table F: The results of crossmodal matching task between RNA and Protein. We report ACC($\uparrow$) as evaluation metric.
>
> ||NeurIPS 2021 Multiomics
> :-|-
> MultiVI|66.32±1.02
> CLUE|74.97±1.53
> Bi2Former(Ours)|84.21±0.48
>
> ### 2). Three or more modalities
> Our framework can be naturally extended to handle three or more modalities with only minor modifications. When multiple modality types (e.g., RNA, ATAC, and Protein) are involved in the graph, we generalize the ABG into a heterogeneous attributed graph. In this setting, each modality is treated as a distinct node type, and edges are constructed to represent biologically meaningful relationships between different modality pairs. During model training, we perform relation-specific attention and aggregation for each edge type. This design enables Bi2Former to support a wide range of tasks including matching, generation, and interaction pattern discovery across any pair of modalities. We evaluated our model on the NeurIPS 2021 Multiomics dataset, the results for the crossmodal matching task are summarized below:
>
> > Table G: The results of crossmodal generation task on the NeurIPS 2021 Multiomics dataset. We report MAE($\downarrow$) as evaluation metric.
>
> ||RNA2ATAC|RNA2Protein
> :-|-|-
> CLUE|0.121|0.147
> Bi2Former(Ours)|0.083|0.102
>
> ***
>
> ## **Q4. About the Negative Sampling Strategy**
> Our method is **robust to the choice of negative sampling ratio**. We have empirically observed that using a positive-to-negative ratio between 1:2 and 2:1 consistently leads to effective training and stable performance.
>
> As illustrated in the table below, the model’s accuracy on both positive and negative samples remains stable across a range of sampling ratios. This indicates that the learned interaction patterns are not overly dependent on the sampling strategy and that the model captures meaningful crossmodal relationships.
>
> > Table J: Model performance on ISSAAC-seq dataset under different negative sampling ratios (e.g., 1:1, 1:2, 2:1).
>
> ||ACC|Pos. Precision|Neg. Precision
> :-|-|-|-
> 1:2|84.19±0.49|82.79±0.52|84.56±0.47
> 1:1|84.40±0.48|84.75±0.42|83.17±0.51
> 2:1|84.06±0.62|84.91±0.69|82.98±0.61
>
> We will include this analysis in Appendix in the final version to clarify the robustness of our sampling strategy.
>
> ***
> Thanks again for your effort in reviewing our work. If you have any additional concerns or suggestions, please feel free to share them, and we sincerely value your feedback.
>
> Sincerely,

---

> > ### Comment · Reviewer_7QNJ · 2025-08-05
> >
> > Thank you to the authors for the rebuttal, which addressed my questions. I suggest revising the presentation of the method to emphasize its ability to effectively handle sparsely paired datasets, as this is a significant challenge in biological measurement data. This point was raised by nearly all reviewers. Although the method outperforms competitors, as demonstrated by the new experiments conducted by the authors with a low number of paired omics samples, highlighting this aspect more clearly would further strengthen the paper. I’ve decided to maintain my initial positive score.

---

> ### Author Response · Authors · 2025-08-07
> **Thank you**
>
> Dear Reviewer 7QNJ,
>
> We sincerely appreciate your positive feedback on our paper.
>
> In the revised version, we have made the following enhancements:
>
> * In the **Introduction**, we clarified our motivation for introducing the modality matching task, explaining its role as a proxy and its widespread use as a benchmark in the field.
>
> * In the **Introduction's summary of contributions**, we further elaborated on the practical utility of our method, **especially in scenarios with sparse or limited data availability** (sparsely paired datasets).
>
> * In **Section 3.1**, we provided additional clarification on the **definition, origin, and significance** of the task setting to better contextualize our work.
>
> * In **Section 4**, we incorporated new experimental results mentioned in the rebuttal, including additional **baselines**, **few-shot**, and **zero-shot** evaluations, and we discussed their implications in detail. Thus **highlighting our ability to effectively handle sparsely paired datasets**.
>
> We will continue to **expand the discussion of related work** to **better position our contribution** within the broader literature.
>
> We sincerely appreciate your efforts; your suggestions have greatly helped us improve this paper. Thank you once again for your encouragement.
>
> Sincerely,

---

### Official Review · Reviewer_e9JF · 2025-07-03

**Clarity:** 3
**Significance:** 2
**Originality:** 2
**Rating:** 3
**Confidence:** 4

**Summary:**

This paper introduces a method based on ABG to match cells with different modalities. The paper is written well over. However, it primarily focuses on an artificial scenario created by the authors, namely, all the datasets they analyzed are from multiomes where the cells already have data from both RNA and ATAC modalities. There is no need to match cells. In this case, I am unsure whether it is a fair comparison with other methods designed for different tasks.

**Questions:**

The authors need to consider the case where the data are unpaired and with methods designed to match cells measured with one modality only.

**Ethical Concerns:**

["NO or VERY MINOR ethics concerns only"]

**Final Justification:**

I will keep my score.

**Limitations:**

The limitations discussed by the authors were not at the core of the problem, namely the artificial nature of their data set-up where the cell are already matched by the experimental design. In this setting, there is very limited utility of their method. Please also see my questions above.

**Paper Formatting Concerns:**

No concerns.

**Quality:**

3

**Strengths And Weaknesses:**

Strengths: The paper is written clearly and the results are well presented Weaknesses: The method is mostly designed for unpaired data, whereas all the data considered are already paired. It is not a fair comparison with other methods designed for paired data for different tasks. I have several questions and I hope the authors can answer them.

1. The motivation is unclear, as the data should be unpaired data and the baselines are all designed for paired data. Including baselines such as scGLUE and scJoint will be helpful.

2. Integration multi-omic data is already well-discussed and analyzed. Could the authors show more biology-driven discoveries to support their method?

3. We do not find examples of tf-binding site analysis in the main text, if the authors intend to argue it as a downstream task, they should include analysis.

---

> ### Author Rebuttal · Authors · 2025-07-31
>
> Dear Reviewer e9JF，
>
> Thank you for your valuable feedback. Below we will address your concerns.
>
> ***
>
> ## **W1 & Q1 Regarding Motivation and Experimental Setting**
> ### 1). Clarification about the "paired" data and the experimental setting
> Thank you for your feedback and valuable suggestions. First, we would like to clarify a possible misunderstanding regarding the experimental setting. Our training setup includes both paired and unpaired data, where unpaired data are generated via negative sampling, consistent with the definition in GLUE documentation[1]:
>
> *“Unpaired” means that different omics layers are not probed in the same single cells, but rather independent samples of (presumably) the same cell population.*
>
> Given a sets of ATAC-RNA instances, where some instances are paired and others are not, our modality matching task is designed to determine whether a instance is paired (originates from the same cell). Importantly, in the testing set, the pairing status lables are **unknown**, meaning the evaluation is effectively conducted on data which not derived from any experimental pairing, making the comparison fair and unbiased.
>
> ### 2). Our core motivation
> Second, we emphasize that the core motivation of our work is not simply to integrate modalities for representation learning (which is only a small part of our contributions), but rather to **leverage crossmodal matching as a proxy** for learning the underlying biological interaction patterns. This yields a model with strong generalization capabilities and biological interpretability, which goes beyond conventional multimodal integration tasks. For a more detailed explanation, please refer to our extended discussion in the response to W1&W2 of Reviewer htWu.
>
> ### 3). Fair comparison and more baselines
> Furthermore, all of the baselines we compared—such as crossmodal generation or representation integration models based on variational autoencoders (VAEs)—also require paired data for training. Their primary motivation is to learn integrated representations for downstream tasks. To ensure a **fair comparison**, we adopt modality matching as a downstream task for all baselines, generating representations from separate modalities and using a classifier head to predict whether two representations are paired. To further alleviate your concern, we have additionally included scGLUE[1] and scJoint[2] as baselines in our updated experiments. The results are shown in the table below:
>
> > Table H: The results of more baselines of crossmodal matching task. We report ACC($\uparrow$) as evaluation metric. Our model consistently achieves superior performance across all evaluated datasets.
>
> ||ISSAAC-seq|10xMultiome-PBMC|SHARE-seq|SNARE-seq|NeurIPS 2021 Multiomics|Avg.
> :-|:-:|:-:|:-:|:-:|:-:|:-:
> scMoGNN|73.72±0.96|72.41±1.37|69.84±1.81|69.03±1.22|75.49±1.31|72.10
> scGLUE|74.28±0.91|72.51±1.02|66.89±1.43|64.47±1.22|76.93±0.82|71.01
> scJoint|72.79±0.85|75.38±0.57|64.32±1.22|67.31±1.39|74.08±0.69|70.77
> Bi2Former(Ours)|84.40±0.48|88.74±0.36|79.84±0.29|73.56±0.37|90.41±0.24|83.39
>
> [1] Multi-omics single-cell data integration and regulatory inference with graph-linked embedding. Nature Biotechnology 2022
>
> [2] scJoint integrates atlas-scale single-cell RNA-seq and ATAC-seq data with transfer learning. Nature Biotechnology 2022
>
> ***
>
> ## **W2 & W3 Biology-driven Discoveries**
> Thank you for your feedback and we apologize for any confusion. We would like to clarify that we have included three types of biology-driven analyses in the main text and appendix. However, due to space limitations, two of them (b and c) are currently located in the appendix, with a brief summary provided in Section 5.5 (Line 308-323) of the main paper. We will expand these discussions in main text in the revised version. A summary of the biology-driven discoveries is provided below:
>
> ### a). Cross-cell-type similarity of attention matrices (In Section 5.3)
> To validate that our learned attention matrices reflect biologically plausible ATAC–RNA regulatory relationships, we leveraged the biological intuition that different cell types should exhibit distinct interaction patterns, and the degree of dissimilarity should correlate with cell-type divergence. We systematically analyzed and compared the pairwise similarity of learned attention matrices across:
> - Different splits of the same cell type.
> - Different cell types within the same dataset.
> - Different cell types across different datasets.
>
> These analyses, summarized in Figure 3 and discussed in Lines 252–263, reveal that attention similarities decline as cell-type differences increase, supporting our hypothesis that Bi2Former captures cell-type-specific regulatory programs.
>
> ### b). Cell-level attention visualization (In Appendix C.1)
>
> To further support the biological plausibility of the learned attention matrices, we visualized the attention weights between ATAC and RNA nodes for individual cells. A known biological principle is that each gene is typically regulated by a limited number of accessible chromatin regions (ATAC peaks), often located on the same chromosome.
>
> In Appendix C.1 (Lines 527–539, Figures 5 & 6), we show that for each RNA, only 5–15 ATAC peaks receive non-negligible attention weights, and these ATACs tend to be clustered by chromosome, aligning with known cis-regulatory mechanisms in gene regulation. These findings reinforce the interpretability and biological grounding of our model’s attention mechanism.
>
> ### c). TF binding site alignment analysis (In Appendix C.2)
>
> We further examined whether the attention scores learned by our model correlate with experimentally derived transcription factor (TF) binding sites, which represent real regulatory activity. Specifically, in Appendix C.2 (Lines 539–556, Figures 7 & 8), we aggregated attention weights across ATAC peaks for CD4 and CD14 cells and compared them with TF binding signals known to be specific to those cell types.
>
> The results show that:
> - In CD4 cells, ATAC peaks with high attention weights from our model coincide with strong CD4-specific TF binding signals.
> - In contrast, CD14 cells show distinct attention patterns, differing from the CD4-specific TF ground truth.
>
> This suggests that our model captures cell-type-specific regulatory mechanisms and aligns well with experimental regulatory evidence, further validating its biological relevance.
>
> ***
>
> ## **L1 About the Utility of Our Method**
> ### 1). Clarification of the experimental setting
> As clarified in our response to Q1, in the testing set, the pairing status lables are unseen for models, meaning the inference is effectively conducted on both **"matched" data** and **unmatched data**. Moreover, our model can determine whether unpaired modalities belong to a same cell without fine-tuning even in zero-shot settings (please check the response to Q1 of Reviewer htWu), which is hard for previous VAE methods, thereby further **reducing the cost of experimental measurements**. Our model is designed to be **highly data-efficient and generalizable**. By incorporating biological attributes as node representations and explicitly modeling crossmodal interaction patterns, our method requires fewer paired samples than previous approaches while achieving superior performance and strong generalization. The results of the performence in different training sizes are detailed in our response to W1&Q1.1&Q1.2 of Reviewer 7QNJ.
>
> ### 2). Crossmodal generation and uncovering crossmodal interaction patterns
> The model also supports **crossmodal generation**, thus further reducing experimental costs. For instance, given a set of ATAC nodes expressed within a cell, the model computes attention scores between these nodes and all RNA nodes in a reference database. By applying a thresholding strategy to the attention matrix, we infer which RNAs are likely to be expressed—effectively generating the RNA modality from the ATAC modality. Compared to traditional VAE-based generative architectures, our generation mechanism offers greater biological intuition and interpretability, while also demonstrating superior performance. The results of partial generation task are shown in the table below:
>
> > Table A: The results of generation task from ATAC modality to RNA modality. We report MAE($\downarrow$) as evaluation metric.
>
> ||ISSAAC-seq|10xMultiome-PBMC|SHARE-seq|SNARE-seq
> :-|:-:|:-:|:-:|:-:
> MultiVI|0.097|0.132|0.114|0.129
> GLUE|0.108|0.140|0.107|0.117
> scMoGNN|0.095|0.101|0.098|0.112
> Bi2Former(Ours)|0.072|0.082|0.079|0.078
>
> Beyond crossmodal matching and generation, the core utility of our method lies in its ability to uncover and interpret fine-grained crossmodal regulatory interactions. By modeling these interactions at a fine scale, our framework supports interpretable crossmodal analysis, enabling new biological insights into cell-type-specific regulatory programs. As discussed in the response to W2&W3.
>
> ### 3). Generalization to more modalities
> Moreover, our framework is **not limited to scRNA-seq and scATAC-seq**. The Bi2Former architecture and its biological-driven attention mechanism are general and can be directly applied to other modality pairs (e.g., RNA–protein, ATAC–methylation) for tasks such as matching, generation, or interpretability with only minor modifications. More analysis and results are included in our response to W2&Q3 of Reviewer 7QNJ.
>
> In summary, the utility of our method is multi-faceted—it not only learns high-quality representations but also reduces experimental cost for matching unpaired data, enables interpretable crossmodal generation, and reveals biological interaction patterns across modalities. In the revised version, we will expand the relevant discussion in the Limitation section.
>
> ***
> Thanks again for your effort in reviewing our work. We hope our response helps to address your concerns. If you still have questions, please don't hesitate to let us know. We’re happy to discuss further.
>
> Sincerely,

---

> ### Comment · Reviewer_e9JF · 2025-08-04
> **Thank you**
>
> Thank you for your responses. I will maintain my scores, as this task is still not well-defined, especially in single-cell data analysis, considering "matching the modality" for each cell is useless due to noise level.

---

> ### Author Response · Authors · 2025-08-05
> **Further Discussion**
>
> Dear Reviewer e9JF,
>
> We sincerely appreciate your response. We are happy to continue discussion. We hope to address your further concerns on the task.
>
> ***
> ### **1) Broad Interest in the Task of "matching the modality"**
>
> Single-cell multi-omics is a rapidly growing research area. Despite the development of joint profiling platforms, **single-modality datasets remain far more prevalent**. A major challenge lies in how to effectively integrate complementary information from multimodal data while leveraging the vast amount of single-modality data to understand cellular states and dynamics.
>
> To address these challenges, [1] categorizes the field into three major tasks:
> (1) Modality prediction,
> (2) Modality matching \[2],
> (3) Joint embedding.
>
> Among these, **modality matching** is a widely acknowledged and actively studied benchmark task [4–6]. It has been the focus of major competitions, including the official NeurIPS Competition [3], and is commonly used to evaluate the performance of new multimodal integration models.
>
> ### **2) Our Task is Practically Aligned with the Standard Setting**
>
> The standard setting of this task is to identify the correspondence between two single-cell profiles and provide the probability distribution of these predictions. Formally:
>
> > Given modality inputs $M_1 \in \mathbb{R}^{N \times k_1}$ and $M_2 \in \mathbb{R}^{N \times k_2}$, the model learns two mapping functions $f_{\theta_1}$ and $f_{\theta_2}$ to project them into a shared space. A scoring function $g$ computes a correspondence matrix
>
> $$
> S = g(f_{\theta_1}(M_1), f_{\theta_2}(M_2)) \in \mathbb{R}^{N \times N}
> $$
> > where $S_{ij}$ represents the predicted probability that cell *i* in modality 1 matches cell *j* in modality 2. The final evaluation score sums the probabilities assigned to the correct matchings.
>
> In our work, we adopt this task to **learn fine-grained interaction patterns between RNA and ATAC nodes**. To facilitate this, we made a slight adaptation to the evaluation: instead of evaluating accuracy based on aggregate soft probabilities over sets, we compute **hard accuracy for each cell’s top-1 prediction**. Nonetheless, the core objective remains the same—**determining whether two modalities originate from the same cell**. Thus, our task is conceptually and practically aligned with the widely accepted definition of modality matching.
>
> To reduce potential misunderstanding, we have further clarified the definition, origin, and significance of the task in Section 3.1 in the revised version.
>
> ### **3) Practical Impact of Our Task and Model**
>
> We would also like to emphasize that **modality matching is a means to an end**, not the end itself in our study.
>
> Our model demonstrates strong generalization across **cell types** and **datasets**, and is capable of **few-shot and zero-shot learning** under extremely sparse paired supervision, which as Reviewer 7QNJ commented "`address a significant challenge in biological measurement data`".
>
> Moreover, our model successfully learns **fine-grained representations** at the RNA, ATAC, and protein levels, and captures **crossmodal interaction patterns** that **align with biological ground truths**. These properties are essential for enabling **biology-driven discoveries and improving interpretability**.
>
> In summary, our method not only achieves state-of-the-art performance on the well-established **modality matching** task, but also demonstrates:
>
> * strong **generalization**,
> * **few-shot** and **zero-shot** learning capability,
> * powerful **generation abilities**,
> * strong **alignment with biological knowledge**, and
> * the abilities of **biology-driven discoveries and improving interpretability**.
>
>  which are all crucial for advancing research in single-cell omics.
>
> To reduce potential misunderstanding, we have revised certain claims in the Introduction to emphasize that modality matching is a means to an end, thereby better highlighting the broader utility of our model.
>
> ***
>
> We appreciate your effort in reviewing our work. We hope our response helps to further address your concerns. If you still have questions, please don't hesitate to let us know. We’re happy to discuss further.
>
> Refence:
>
> [1] A sandbox for prediction and integration of DNA, RNA, and proteins in single cells. NeurIPS 2021
>
> [2] MATCHER: manifold alignment reveals correspondence between single cell transcriptome and epigenome dynamics. Genome biology 18
>
> [3] https://openproblems.bio/events/2021-09_neurips
>
> [4] Cross-Linked Unified Embedding for cross-modality representation learning. NeurIPS 2022
>
> [5] MultiVI: deep generative model for the integration of multimodal data. Nature Methods 2023
>
> [6] scMaui:a widely applicable deep learning framework for single-cell multiomics integration in the presence of batch effects and missing data. BMC bioinformatics 25

---

> > ### Author Response · Authors · 2025-08-07
> > **Supplements for Further Discussion**
> >
> > To further demonstrate that our task is **fundamentally aligned with the widely studied and benchmarked standard modality matching setting**, we report comparative performance evaluations under both our task formulation (*accuracy-based*) and the standard formulation (*matching score-based*).
> >
> > Specifically, we evaluate performance on two datasets—**ISSAAC-seq** and the **NeurIPS 2021 competition** dataset—and compute **correlation metrics** between the two sets of results. The findings reveal a strong **positive correlation**, supporting that the two task formulations are **intrinsically consistent**.
> >
> > ---
> >
> > > Table : Performances for modality matching on the **ISSAAC-seq** dataset under our task setting (accuracy) and the standard setting (matching score).
> >
> > | |Ours (ACC$\uparrow$)|Standard (Matching Score$\uparrow$)
> > :-|:-:|:-:
> > MultiVI|66.21|0.0104
> > CLUE|74.28| 0.0610
> > scMoGNN|73.72| 0.0587
> > MLP |67.39|0.0122
> > Bi2Former|84.40|0.0921
> >
> >
> > * **Pearson Correlation Coefficient**: **0.970**
> > * **Mutual Information**: **0.922** *(normalized)*
> >
> > These metrics demonstrate a **strong linear correlation** and **shared information structure** between our task and the standard benchmark.
> >
> > ---
> >
> > > Table : Performances for modality matching on the **NeurIPS 2021 Competition** dataset under our task setting (accuracy) and the standard setting (matching score).
> >
> > | |Ours (ACC$\uparrow$)|Standard (Matching Score$\uparrow$)
> > :-|:-:|:-:
> > MultiVI|67.44|0.0113
> > CLUE|73.21| 0.0583
> > scMoGNN|75.49| 0.0630
> > MLP |65.21|0.0095
> > Bi2Former|90.41|0.0987
> >
> > * **Pearson Correlation Coefficient**: **0.952**
> > * **Mutual Information**: **0.904** *(normalized)*
> >
> > Once again, the results confirm that **performance trends across models remain highly consistent** between the two task definitions, further validating that our task setup adheres to the recognized paradigm of modality matching.

---

> > > ### Author Response · Authors · 2025-08-08
> > > **Kind Reminder of the Discussion Period Ending Soon**
> > >
> > > Dear Reviewer e9JF,
> > >
> > > We are writing to kindly remind you that the discussion period will conclude on Aug 8, 11.59pm AoE. We would like to know whether our further discussion has addressed your remaining concern, and we would really appreciate it if you could take some time to read our responses and provide any further feedback that you have. Your feedback is important to us, and we would like to have the opportunity to discuss them with you before the discussion period ends.
> > >
> > > Thank you again for helping us, we believe our paper is standing much stronger after incorporating your constructive feedback.
> > >
> > > Best regards,

---

> > > > ### Comment · Area_Chair_cPQ5 · 2025-08-09
> > > >
> > > > Reviewer e9JF,
> > > >
> > > > Have your concerns be addressed by the authors?
> > > > What is your assessment of the rebuttal?
> > > >
> > > > Best,
> > > > the AC

---

### Official Review · Reviewer_KGd3 · 2025-07-04

**Clarity:** 2
**Significance:** 2
**Originality:** 2
**Rating:** 4
**Confidence:** 2

**Summary:**

The paper presents  Bi$^2$ Former - a method for cross-modal matching between scRNA and ATAC data and predicting whether the data originates from the same cell or not. In order to do this, the authors leverage an attributed bipartite graph (with explicit features for scRNA data (in addition to gene expression) and ATAC data (in addition to chromatin accessibility)) and learn an attention-based mechanism for inferring cross-modal matching. They leverage domain-informed adjacency bipartite graph as well as biological pruning to eliminate noise and get a robust signal.

The authors show with evaluation on 4 datasets  that their model performs better than state-of-the-art for crossmodal matching. They also show ablation study highlighting contribution of different features including integration of additional node attributes, influence of domain-informed base adjaceny matrix, biological pruning and (learnt) node embeddings.

**Questions:**

- In the ablation study, the influence of Edge is highest. I understand the chromosomal mask as $A_{ij}=1$ defining whether gene $i$ lies on chromosome $j$ or $0$ if not. In the ablation without the intrachromosomal restriction, what is the fraction of inter-chromosomal interactions post biological pruning i.e., $ \alpha_{ij}=1$ where $ A_{ij}=0 $ ?
- How could the method be extended to more than two modalities?

**Ethical Concerns:**

["NO or VERY MINOR ethics concerns only"]

**Final Justification:**

Changed to weak accept after review discussion

**Limitations:**

yes

**Quality:**

2

**Strengths And Weaknesses:**

# Strengths

- Paper is well-written.
- Strong empirical evidence is provided on the utility of the method.

# Weaknesses
- MultiModal data integration for scRNA-seq data and scATAC data is an important problem. However, there are now kits available, that allow collection of both these modalities from the same cell (e.g., https://www.10xgenomics.com/products/epi-multiome). Further, several methods have been developed for such aligned data including Bayesian approaches like MOFA (https://www.embopress.org/doi/full/10.15252/msb.20178124), MOFA+(https://genomebiology.biomedcentral.com/articles/10.1186/s13059-020-02015-1). Other approaches including linear algebraic or deep learning based are also relevant. Please see review in https://arxiv.org/abs/2501.17729 .

My first question: The paper should clearly position its contribution compared to existing approaches. Specifically, if only targetting scATAC-seq and scRNAseq, since there are technologies that make the whole problem moot  by providing both modalities for the same cell (when using 10X kits), the significance should be highlighted.

Also, would it be possible to show the results of the method on 3k cells from the frozen human healthy brain tissue on data by 10x Genomics here: https://www.10xgenomics.com/resources/datasets/frozen-human-healthy-brain-tissue-3-k-1-standard-2-0-0

- Alternative modalities for which the data has to necessarily be matched are present, e.g.,  scRNAseq with Cut&Tag methods that can provide high-resolution signal compared to ChiPseq used in the past. How can the method be extended beyond scATACseq+snRNAseq integration?

- I have not worked with scATAC data myself but am very familiar with previous single cell pipelines (e.g., SEURAT, scanpy, etc.). These pipelines are well-established in the field. For example, see https://satijalab.org/seurat/articles/seurat5_atacseq_integration_vignette . What would be the practicality of this method compared to previous methods.

---

> ### Author Rebuttal · Authors · 2025-07-31
>
> Dear Reviewer KGd3,
>
> Thanks for your comments. Below we will address your questions.
>
> ***
>
> ## **W1 The Contribution (or Motivation & Utility) of the Work**
> We sincerely apologize for any confusion, and we would like to clarify our contributions and utility from three perspectives:
>
> ### 1). Reducing Experimental Measurement Costs
> While technologies such as 10X allow for profiling both modalities (e.g., RNA and ATAC) from the same cell, they require destructive and costly protocols, making the acquisition of paired multi-omics data **expensive and limited**.
> Importantly, our method is **not in conflict with such technologies**; instead, it leverages a small amount of paired data obtained via such technologies to train a model that generalizes well to unseen data. Our model is designed to be **highly data-efficient and generalizable**, and thus can significantly reduce the cost for paired data in the following scenarios:
> - **Semi-paired setting**: When only a subset of cells or cell types have paired measurements, our model can be trained on this limited data and generalize to unseen cells by leveraging biological attributes and interaction patterns learned during training.
> - **Unpaired setting (zero-shot)**: Existing VAE-based models typically require retraining or fine-tuning on new datasets. Our model can be pretrained on available source datasets and directly evaluated on target datasets without any fine-tuning. This simulates a real-world use case where no pairing is available at test time. Results from our zero-shot experiments are presented in the response to Q1 of Reviewer htWu.
> - **Crossmodal generation**: Beyond matching, our model also supports crossmodal generation, i.e., predicting one modality from another, even under the zero-shot setting. This allows estimation of unmeasured modalities, further reducing experimental costs. Compared to traditional VAE-based generative architectures, our generation mechanism offers greater biological intuition and interpretability, while also demonstrating superior performance. The pipeline and results of this generation task are detailed in our response to L1.2 of Reviewer e9JF.
>
> ### 2). Uncovering Fine-Grained Crossmodal Interactions
> Current biological measurement techniques and prior deep learning-based models for crossmodal integration (or matching and generation) **fail to explicitly model or learn fine-grained interactions between modalities** (e.g., how chromatin accessibility influences gene expression). In contrast, the core utility of our approach lies not merely in performing matching, integration, or generation tasks(which is only a small part of our contributions), but rather in leveraging crossmodal matching as a proxy for **learning and uncovering the underlying biological interaction patterns** which are highly aligned with known biology. This leads to strong generalization, interpretability, and biological plausibility. We provide extensive empirical validation of the biological relevance, interpretability, and discovery potential of the learned attention patterns in our work. These results are discussed in detail in the response to W2&W3 of Reviewer e9JF.
>
> ### 3). Generalization to Other Modalities
> Our framework is **not limited to scRNA-seq and scATAC-seq**. The Bi²Former architecture and its biological-driven attention mechanism are general and can be directly applied to other modality pairs (e.g., RNA–protein, ATAC–methylation) for tasks such as matching, generation, or interpretability with only minor modifications. More analysis and results are included in the response to Q2 below.
>
> Overall, while existing experimental technologies and modeling frameworks (e.g., Bayesian or deep learning-based integration) have enabled multi-omics analysis, our study **addresses distinct and previously underexplored challenges**.
>
>
> ***
>
> ## **W2 Results on Flash-Frozen Human Healthy Brain Tissue (3k) Dataset**
>
> We have integrated the Flash-Frozen Human Healthy Brain Tissue (3k) dataset in our experiments. The results are shown in the table below:
>
> > Table D: The results of crossmodal matching task and generation from ATAC modality to RNA modality on Flash-Frozen Human Healthy Brain Tissue (3k) dataset.
>
> ||Matching(ACC($\uparrow$))|Generation(MAE($\downarrow$))
> :-|:-:|:-:
> MultiVI|72.13|0.124
> CLUE|74.89|0.107
> scMoGNN|76.84|0.118
> Bi2Former(Ours)|95.79|0.069
>
> We noticed that chromosome information is not available in this dataset, so we constructed the graphs without edge connections. Nevertheless, Bi2Former still demonstrated strong performance, further highlighting the robustness of our approach.
>
> ***
>
> ## **W3 Extended beyond scATACseq+scRNAseq integration**
> Please check the response to Q2 below.
>
> ## **W4 The Practicality Ccompared to Previous Methods**
> Please refer to the response to W1.
>
> ***
>
> ## **Q1 The Fraction of Inter-chromosomal Interactions**
> In our framework, the graph edges serve multiple purposes:
> - They introduce biological priors into the Attributed Bipartite Graph (ABG) to enhance learning performance;
> - They provide inductive bias to guide the message aggregation process, improving training efficiency and convergence;
> - They help reduce computational complexity by constraining attention computation to the existing sparse edges.
>
> As analyzed in Section 5.4 (Line 279–285), when edges are removed, the model's performance degrades but still remains competitive compared to VAE-based baselines. To further address your concern, we conducted additional experiments to evaluate the learned attention under a noisy topology (without edges):
>
> > Table E: The results of the fraction of inter-chromosomal interactions post biological pruning on ISSAAC-seq dataset.
>
> ||ACC($\uparrow$)|Inter-chromosomal(%)
> :-|:-:|:-:
> MultiVI| 66.21| -
> CLUE|71.28|-
> Bi2Former|84.40|100.00
> Bi2Former wo. Edge|78.89|94.87
>
> These results demonstrate that our attention mechanism remains effective even without edge constraints. Notably, the Bi2Former without predefined edges still learns interaction patterns that are predominantly intrachromosomal, aligning with biological expectations. This highlights the model’s robustness and its capacity to infer biologically meaningful interactions even under edge-free or noisy topologies.
>
> ***
>
> ## **Q2 Generalization to Other Modalities**
> ### 1). Two modalities
> Our framework is **not limited to scRNA-seq and scATAC-seq**. The Bi²Former architecture and its biological-driven attention mechanism are general and can be directly applied to other modality pairs (e.g., RNA–Protein, ATAC–Methylation) for tasks such as matching, generation, or interpretability with only minor modifications.
> Specifically, the only modifications needed are in the biological attributes of nodes and the edge construction strategy, depending on the modalities involved. The core model architecture remains unchanged, showcasing the flexibility and extensibility of our approach. To further address your concern, we also empirically report the RNA-Protein matching results:
>
> > Table F: The results of crossmodal matching task between RNA and Protein. We report ACC($\uparrow$) as evaluation metric.
>
> ||NeurIPS 2021 Multiomics
> :-|-
> MultiVI|66.32±1.02
> CLUE|74.97±1.53
> Bi2Former(Ours)|84.21±0.48
>
> ### 2). Three or more modalities
> Our framework can be naturally extended to handle three or more modalities with only minor modifications. When multiple modality types (e.g., RNA, ATAC, and Protein) are involved in the graph, we generalize the ABG into a heterogeneous attributed graph. In this setting, each modality is treated as a distinct node type, and edges are constructed to represent biologically meaningful relationships between different modality pairs. During model training, we perform relation-specific attention and aggregation for each edge type. This design enables Bi2Former to support a wide range of tasks including matching, generation, and interaction pattern discovery across any pair of modalities. We evaluated our model on the NeurIPS 2021 Multiomics dataset, the results for the crossmodal matching task are summarized below:
>
> > Table G: The results of crossmodal generation task on the NeurIPS 2021 Multiomics dataset. We report MAE($\downarrow$) as evaluation metric.
>
> ||RNA2ATAC|RNA2Protein
> :-|-|-
> CLUE|0.121|0.147
> Bi2Former(Ours)|0.083|0.102
>
> ***
> Thanks again for your comments. If you have any additional concerns or suggestions, please feel free to share them, and we sincerely value your feedback.
>
> Sincerely,

---

> > ### Comment · Reviewer_KGd3 · 2025-08-06
> >
> > Dear authors, Thank you for answering my queries. I am revising my score to "4: weak accept" given the discussion. However, I believe the paper needs some rewriting with better exposition.

---

> ### Author Response · Authors · 2025-08-07
> **Thank you**
>
> Dear Reviewer KGd3,
>
> We sincerely appreciate your positive feedback on our paper. We are also grateful for your constructive suggestions, which have helped us improve the manuscript.
>
> In the revised version, we have made the following enhancements:
>
> * In the **Introduction**, we clarified our motivation for introducing the modality matching task, explaining its role as a proxy and its widespread use as a benchmark in the field.
>
> * In the **Introduction's summary of contributions**, we further elaborated on the practical utility of our method, especially in scenarios with sparse or limited data availability.
>
> * In **Section 3.1**, we provided additional clarification on the **definition, origin, and significance** of the task setting to better contextualize our work.
>
> * In **Section 4**, we incorporated new experimental results mentioned in the rebuttal, including additional **baselines**, **few-shot**, and **zero-shot** evaluations, and we discussed their implications in detail.
>
> We will continue to **expand the discussion of related work** to **better position our contribution** within the broader literature.
>
> Thank you once again for your support.
>
> Sincerely,

---

### Official Review · Reviewer_htWu · 2025-07-05

**Clarity:** 2
**Significance:** 2
**Originality:** 1
**Rating:** 4
**Confidence:** 5

**Summary:**

This work introduces Bi²Former, a biologically-informed bipartite graph transformer designed for single-cell cross-modal matching. The problem is framed as a graph classification task using Attributed Bipartite Graphs (ABGs), enabling explicit modeling of interactions between ATAC and RNA features. Bi²Former learns interpretable attention over regulatory pairs to uncover cross-modal relationships. Extensive experiments demonstrate state-of-the-art performance, strong generalization to unseen cell types, and biologically meaningful insights. The proposed framework offers an interpretable and structured approach to cross-modal learning in single-cell biology.

**Questions:**

For Cross-Cell-Type Generalization, can you train on one dataset to evaluate on a separate dataset with unseen cell types in training?

**Ethical Concerns:**

["NO or VERY MINOR ethics concerns only"]

**Final Justification:**

finer-grained modeling paradigm for graph neural networks is introduced in single-cell multi-omics, introducing attributed bipartite graphs that shift focus from cell-level to gene- and peak-level representation learning, enabling more biologically meaningful and generalizable interaction capture.

**Limitations:**

The tarining is based onthe paired scRNA-seq and scATAC-seq data, which is limited.

**Quality:**

2

**Strengths And Weaknesses:**

Strengths:
1. The work is easy to follow.

Weaknesses:
1. The proposed method is not novel to me.
2. highly relevant and similar work missing, scMoGNN
Since your method also uses a bipartite-graph formulation for modality matching, please compare it with scMoGNN—to my knowledge the first GNN-based approach for multi-omics prediction, modality matching, and joint embedding—which likewise builds a bipartite graph.
a. Bipartite graphs for multi-omics integration, ATAC-RNA, RNA-protein, and so on.
b. also including prior knowledge
paper: Graph Neural Networks for Multimodal Single-Cell Data Integration, KDD 2022, NeurIPS Multiomics Integration Winner solution
3. A highly relevant baseline is missing, scMoGNN; please also include the NeurIPS 2021/2022 Multiomics integration dataset, which is larger.

---

> ### Author Rebuttal · Authors · 2025-07-31
>
> Dear Reviewer htWu，
>
> Thanks for your constructive comments. Below we will address your concerns.
>
> ***
>
> ## **W1&W2. Regarding Novelty and Similar Work**
>
> We sincerely thank you for raising this important point. However, we believe there is a misunderstanding. While scMoGNN also utilize graph neural networks for modeling single-cell multi-omics data, our methodology and contributions differ significantly from prior works in terms of **motivation, graph construction, model design, and task definition**.
>
> ### 1. Motivation
> scMoGNN is primarily designed to learn integrated cell-level representations by modeling the cells and their omics features (e.g., RNA, ATAC) through a graph. It leverages GNNs for aggregation at the **inter-cell level**, yielding **integrated embeddings**.
>
> In contrast, our motivation is to **uncover fine-grained intra-cell crossmodal interactions**(e.g., how chromatin accessibility influences gene expression), aiming for a model that is highly generalizable and biologically interpretable. To achieve this, we emphasize the **node-level** modeling of pairwise interactions between expressed RNA and ATAC within each cell as an independent interaction instance, enabling the model to capture the interaction patterns between them (Or any two or among more modalities).
>
> ### 2. Graph Construction
> scMoGNN constructs a graph where **each cell is treated as a node**, and omics features (e.g., ATAC, RNA) are also represented as nodes. It is important to note that in scMoGNN, both the cell nodes and feature nodes (i.e., RNA and ATAC nodes) do not carry biological attributes as learnable features—thus it is a **non-attributed Bipartite Graph**, which prevents the model from learning generalizable properties from biological attributes (Leading challenge 2 in our paper). Moreover, the **reliance on predefined prior graphs** limits scalability and generalization across datasets, and prevents it from capturing fine-grained crossmodal interaction patterns, leading to a lack of interpretability (Suffering challenge 3).
>
> In contrast, our model builds an **Attributed Bipartite Graph** (ABG) for each individual cell, in which RNA and ATAC features are treated as nodes enriched with biologically meaningful attributes. These attributes enable the model to learn representations that generalize across datasets and cell types. Additionally, our edge construction leverages biological priors in a dynamic and cell-specific manner—e.g., connecting ATAC and RNA nodes based on chromatin co-localization—to embed domain priors into the graph, allowing the graph structure to reflect fine-grained, biologically interpretable crossmodal interactions.
>
> ### 3. Model Design
> In line with its graph construction, scMoGNN primarily focuses on aggregating multi-hop neighborhood information to obtain cell-level embeddings using **a basic Graph Convolution Networks**. Its reliance on predefined and limited interaction priors means the model emphasizes leveraging known biological priors, rather than learning or discovering interaction patterns.
>
> In contrast, our model is designed to learn potential crossmodal interaction patterns between RNA and ATAC features (Or any two or among more modalities) through a **biological-driven Graph Transformer**. Instead of using a fixed structure, we allow the model to learn interaction weights from data. The resulting attention maps not only align well with actual biological evidence (e.g., TF-binding signals), but also contributing to strong generalization capability of our model across cell types and datasets.
>
> ### 4. Formulated Task
> It is important to clarify that although scMoGNN includes a modality matching task, its formulation and purpose differ significantly from ours. Specifically, scMoGNN aims to predict the **probability** that two profiles from different modalities originate from the same cell, serving as an auxiliary task to learning high-quality cell-level representations. Learning the crossmodal interaction patterns remains a challange.
>
> In contrast, our modality matching task serves as a proxy to uncover biological interaction patterns between modalities, as the correct matches reflect the interaction patterns governed by underlying biology. Rather than estimating the probability, our method explicitly models the **binary decision** of whether a given RNA–ATAC pair is a biologically meaningful match. This enables our model to determine whether unpaired modalities belong to a same cell without fine-tuning, thereby reducing the cost of experimental measurements.
>
> Apart from the modality matching task (see the supplemented comparison results in the response to W3), our model also supports **crossmodal generation**. The specific generation pipeline of our method can refer to our response to L1.2 of reviewer e9JF. The results of generation task are shown in the table below:
>
> > Table A: The results of generation task from ATAC modality to RNA modality. We report MAE($\downarrow$) as evaluation metric.
>
> ||ISSAAC-seq|10xMultiome-PBMC|SHARE-seq|SNARE-seq
> :-|:-:|:-:|:-:|:-:
> MultiVI|0.097|0.132|0.114|0.129
> GLUE|0.108|0.140|0.107|0.117
> scMoGNN|0.095|0.101|0.098|0.112
> Bi2Former(Ours)|0.072|0.082|0.079|0.078
>
> Overall, while basic GNNs have been explored in prior work as powerful tools for single-cell multi-omics and many other AI4Science problems, our study **addresses distinct and previously underexplored challenges** (uncovering the nature of crossmodal interaction patterns, achieving strong generalization through biological attributes, and enhancing biological interpretability). We will cite scMoGNN in the related work section and introduce these differences to prevent further misunderstanding.
>
> ***
>
> ## **W3. About the Baseline**
> We appreciate the suggestion to include scMoGNN as a baseline and the NeurIPS dataset. We have integrated scMoGNN and NeurIPS 2021 dataset in our experiments and will include results and discussions in the revised version. The results are shown in the table below:
>
> > Table B: The results of more baselines of crossmodal matching task. We report ACC($\uparrow$) as evaluation metric.
>
> ||ISSAAC-seq|10xMultiome-PBMC|SHARE-seq|SNARE-seq|NeurIPS 2021 Multiomics|Avg.
> :-|:-:|:-:|:-:|:-:|:-:|:-:
> scMoGNN|73.72±0.96|72.41±1.37|69.84±1.81|69.03±1.22|75.49±1.31|72.10
> Bi2Former(Ablation variant)|74.28±1.07|72.54±1.46|70.19±2.06|69.84±1.27|77.83±0.98|72.94
> Bi2Former(Ours)|84.40±0.48|88.74±0.36|79.84±0.29|73.56±0.37|90.41±0.24|83.39
>
> Our model consistently achieves superior performance across all evaluated datasets. More importantly, when comparing against an ablation variant of our model (in which biological attributes and edges are removed, such that the primary difference lies in the model architecture), our method still outperforms scMoGNN. This provides evidence that our biologically-driven attention-based framework is more suitable for capturing interactions in single-cell data than simple GCN aggregation, further highlighting the novelty and significance of our proposed approach.
>
> ***
>
> ## **Q1. Further Generalization Ability Experiments**
> To further demonstrate the robustness and generalization ability of our model, we have conducted zero-shot cross-dataset experiments. Specifically, we pretrained the model on source datasets and evaluated it directly on different target datasets without any fine-tuning. The results are summarized in the table below:
>
> > Table C: Zero-shot inference results for crossmodal matching task with training across various source dataset(s). The target dataset is ISSAAC-seq.
>
> Source dataset(s)|10xMultiome-PBMC|10xMultiome-PBMC+SHARE-seq|10xMultiome-PBMC+SHARE-seq+SNARE-seq
> :-|:-:|:-:|:-:
> MultiVI|47.82|48.19|51.24
> CLUE|51.07|49.67|52.93
> scMoGNN|54.74|61.02|58.89
> Bi2Former(Ours)|71.14|79.16|80.03
>
> Under the zero-shot setting, our model achieves a clear advantage over the baselines. As the size of the training dataset increases, the performance on the target dataset continues to improve, demonstrating that our model effectively generalizes by learning from biological attribute information and discovering crossmodal interaction patterns.
>
> ***
>
> ## **L1. About the Limitation**
> While it is true that paired data are required during training (which is necessary during training for almost all crossmodal integration models), our model is designed to be **highly data-efficient and generalizable**. By incorporating biological attributes as node representations and explicitly modeling crossmodal interaction patterns, our method requires fewer paired samples than previous approaches while achieving superior performance and strong generalization. The results of the performence in different training sizes are detailed in our response to W1&Q1.1&Q1.2 of Reviewer 7QNJ. In the revised version, we have added concerns about paired data to the limitations section.
>
> Furthermore, our framework is **not limited to scRNA-seq and scATAC-seq**. The Bi²Former architecture and its biological-driven attention mechanism are general and can be directly applied to other modality pairs (e.g., RNA–protein, ATAC–methylation) for tasks such as matching, generation, or interpretability with only minor modifications. More analysis and results are included in our response to W2&Q3 of Reviewer 7QNJ.
>
> ***
> Thanks again for your effort in reviewing our work. If you have any additional concerns or suggestions, please feel free to share them, we would be happy to discuss further. We sincerely value your feedback.
>
> Sincerely,

---

> > ### Comment · Reviewer_htWu · 2025-08-05
> > **Thank you**
> >
> > I will increase the score to 3. But the use of an attributed bipartite graph compared to a non-attributed bipartite graph, as in scMoGNN, does not constitute a significant novelty.

---

> ### Author Response · Authors · 2025-08-05
> **Further Discussion**
>
> Dear Reviewer htWu,
>
> Thanks for your feedback. We hope to address your concerns regarding novelty in a more detailed way. We provide clarification from the following perspectives:
>
> ***
> **Digging into Unsovled Challenges**: While previous studies on single-cell multi-omics include innovative designs which used GNNs, they address different issues and have not adequately explored or resolved our concerns regarding mining fine-grained interaction patterns and learning gene-level and ATAC peak-level biological representations, which can align with biological ground truths and provide strong generalization capabilities. These gaps leave several important issues unresolved.
>
> For example, the goal of GLUE [1] is to leverage prior knowledge of feature interactions to bridge the modality gap during integration of unpaired multi-omics data. It organizes RNA gene and ATAC peak interactions into a guidance graph to direct the integration process. However, it does not support spontaneous interaction learning between modalities. Similarly, scMoGNN [2] also constructs graphs based on prior feature relationships to enhance cell-level representation learning. However, it *lacks the capability to learn gene-level, ATAC peak-level, and protein-level representations*, thereby limiting its generalization *across cell types and datasets*. Moreover, it *fails to align with biological ground truths*, which are essential for enabling *biology-driven discoveries and improving interpretability*.
>
> **Proposing a New Modeling Paradigm**:
> In order to solve the above challenges, we introduce a finer-grained modeling paradigm for graph neural networks in single-cell multi-omics analysis. The concept of an `attributed bipartite graph`, compared to a `non-attributed bipartite graph`, is not merely about adding features to nodes—it represents a shift in motivation, graph construction, and model design. Our approach transitions from cell-level representation learning to gene- and peak-level representation learning, enabling the model to capture interactions that are both biologically meaningful and generalizable.
>
> **Complementarity and Integration with Existing Methods**:
> Our perspective does not conflict with current GNN-based single-cell methods. Instead, it is compatible and can be integrated to enhance these methods further. Specifically, the RNA–ATAC and RNA–protein interaction patterns learned by Bi2Former can serve as guidance graphs or prior knowledge to enhance methods like GLUE and scMoGNN, which rely heavily on pre-defined interactions.
>
> To demonstrate this, we incorporated the RNA–ATAC and RNA–protein interaction matrices learned by Bi2Former into scMoGNN's adjacency matrix as prior knowledge. The results show significant improvement in performance on the joint embedding task. The table below summarizes the results:
>
> > Table: Performances for Joint Embedding. The task settings follow those in the original scMoGNN paper. "Ours" refers to the GEX-ADT and GEX-ATAC interaction pattern matrices learned in matching tasks by Bi2Former, which are then used as priors in the scMoGNN adjacency matrix.
>
> | |NMI cluster/label|Cell type ASW|Cc_con|Traj_con|Batch ASW|Graph connectivity|Average metric
> :-|:-:|:-:|:-:|:-:|:-:|:-:|:-:
> GLUE|0.8022| 0.5759| 0.6058| 0.8591| 0.8800| 0.9506| 0.7789
> scMoGNN|0.8499 ± 0.0032| 0.6496 ± 0.0046| 0.7084 ± 0.0316| 0.8532 ± 0.0019| 0.8691± 0.0020| 0.9708 ± 0.0041| 0.8168
> scMoGNN + Ours|**0.8673 ± 0.0021**| **0.6598 ± 0.0042**| **0.7814 ± 0.0429**| **0.8732 ± 0.0081**| **0.8912± 0.0079**| **0.9829 ± 0.0068**| **0.8426**
>
> In summary, our work identifies and tackles important yet underexplored challenges in single-cell multi-omics modeling—specifically, the need for fine-grained, generalizable, and biologically aligned interaction learning. We propose a new modeling paradigm and demonstrate not only strong performance, but also the ability to enhance existing methods. These results collectively support the novelty and broad applicability of our approach.
>
> ***
> We appreciate your effort in reviewing our work. We hope our response helps to further address your concerns. If you still have questions, please don't hesitate to let us know. We’re happy to discuss further.
>
> [1] Multi-omics single-cell data integration and regulatory inference with graph-linked embedding. Nature Biotechnology 2022
>
> [2] Graph Neural Networks for Multimodal Single-Cell Data Integration. KDD 2022

---

> > ### Author Response · Authors · 2025-08-08
> > **Kind Reminder of the Discussion Period Ending Soon**
> >
> > Dear Reviewer htWu,
> >
> > We are writing to kindly remind you that the discussion period will conclude on Aug 8, 11.59pm AoE. We would like to know whether our further discussion has addressed your concern about novelty, and we would really appreciate it if you could take some time to read our responses and provide any further feedback that you have. Your feedback is important to us, and we would like to have the opportunity to discuss them with you before the discussion period ends. Thank you again for your time and effort.
> >
> > Best regards,

---

> > > ### Comment · Reviewer_htWu · 2025-08-08
> > > **Thank you**
> > >
> > > Thanks to the authors for the clarification. Pending a revision that clearly and explicitly describes the differences between scMoGNN and your method, the paper will be clearer. I am happy to increase my score to 4.

---

> > > > ### Author Response · Authors · 2025-08-09
> > > > **Thank you!**
> > > >
> > > > Dear Reviewer htWu,
> > > >
> > > > We sincerely appreciate your positive feedback on our paper. We have expanded the discussion of scMoGNN in the related work by clarifying their differences to better position this work and alos included the new experimental result in the revised version.
> > > >
> > > > Once again, thank you for your encouragement.
> > > >
> > > > Sincerely,

---

### Note · Authors · 2025-08-12

Dear Reviewers, AC and SAC,

We sincerely appreciate your time and effort in reviewing our paper. To facilitate the AC–reviewer discussion, we have summarized below the key consensus reached during the discussion phase and the corresponding improvements made to the paper.

During the discussion, we primarily addressed points regarding the novelty, experimental settings, and the practical significance of both the task and the proposed model. By adding clarifications and refining the presentation in the revised manuscript, we reduced potential misunderstandings, better positioned our work, and highlighted its significance. As a result, we reached consensus with Reviewers htWu, KGd3, and 7QNJ, who resolved their initial concerns, acknowledged the merits of our work, and gave us positive recommendations.

Specifically, in the revised version we made the following enhancements:

- (1) **Strengthened motivation and contributions**: \
In the Introduction, we further clarified our motivation for introducing the modality matching task, explaining its widespread use as a benchmark in the field and emphasizing its role as a proxy in our work. In the summary of contributions, we further elaborated on the practical utility of our method, especially in scenarios with sparse or limited data availability (sparsely paired datasets).


- (2) **More detailed background description**: \
In the Related Works, we added discussion of relevant studies that apply GNNs to single-cell omics, and better positioned our contribution within the broader literature.
In Section 3.1 (Problem Definition), we provided additional clarification on the definition and significance of the task setting to better contextualize our work.

- (3) **Supplementary robustness experiments**：\
In Section 5 (Experiments), we incorporated new experimental results mentioned in the rebuttal, including additional baselines, few-shot and zero-shot evaluations, and generalization to more modalities, further highlighting our method’s utility, ability to effectively handle sparsely paired datasets, and biology-driven discoveries.

At present, the only negative score and remaining divergence come from Reviewer e9JF’s reservations about the modality matching task. We trust that our further clarification during the rebuttal period will address the reviewer's concerns.

Once again, we appreciate your constructive feedback and encouragement, which have significantly improved our paper.

Sincerely,

Bi2Former Authors

---

### Decision · Program_Chairs · 2025-09-17

**Decision:**

Accept (poster)

**Comment:**

This paper reframes single-cell cross-modal matching as graph classification on Attributed Bipartite Graphs (ABGs) and introduces Bi²Former, a biologically guided bipartite graph transformer that attends over ATAC–RNA pairs to capture putative regulatory interactions. The method delivers strong accuracy across multiple datasets, shows cross-cell-type and cross-dataset generalization (few/zero-shot), extends to RNA–protein and ATAC→RNA generation, and offers interpretable attention patterns consistent with cis-regulation and TF-binding signals. The model’s efficiency is supported by sparse attention on biologically plausible edges and node reduction to expressed features.

Concerns around novelty and task framing were substantively addressed. The authors positioned their contribution relative to scMoGNN and related GNN/transformer baselines, emphasizing per-cell attributed graphs and biologically driven attention that learn reusable interaction patterns; they added the missing baselines (scMoGNN, scGLUE, scJoint), a larger 10x brain 3k dataset, standardized metric comparisons (with high correlation to community matching scores), robustness to negative sampling, and runtime analyses showing competitive or better training time than VAE baselines. Biological validation—cell-type-specific attention structure and TF-binding alignment—was strengthened, and zero-shot/sparse-pairing studies convincingly demonstrate practical utility when paired data are limited.

Reviewer sentiment converged positively after rebuttal: htWu and KGd3 raised to weak-accept following clarifications and added experiments; 7QNJ remained positive, highlighting value in sparsely paired regimes; e9JF maintained reservations about the task’s utility, but did not engage in a discussion with the authors that have argued their case convincingly.